# Split Group Knockoffs: Controlling False Discovery Rate in Transformational Group Sparsity

**Siqi Chen** [1]  **Yachen Gao** [2][3]  **Yanwei Fu** [3][4]  **Xinwei Sun** [4]

## Abstract

Controlling the false discovery rate (FDR) under complex sparsity structures remains a fundamental challenge in large language model (LLM) analysis. Motivated by multiple comparison problems in LLMs, we consider a setting in which sparsity arises at the group level after a linear transformation of model parameters. We propose *Split Group Knockoffs (SGKs)*, a general framework for group-wise variable selection under grouped transformational sparsity that extends the Split Knockoff procedure to grouped transformed variables. We establish theoretical guarantees for group-level FDR control and support recovery consistency, addressing challenges induced by group-wise penalties in transformed spaces. Applying SGK to LLM behavior auditing experiment reveals that model disagreement is not uniform across subjects, but instead concentrates in domains with greater semantic and reasoning complexity, where SGK effectively distinguishes genuine behavioral deviations from surface-level performance variation.

## 1. Introduction

Analyses of large language models (LLMs) typically focus on group-level behavior, rather than element-wise responses. Such grouping arises at multiple levels: at the architectural level, parameters are organized into components such as layers and attention heads, each responsible for distinct computational roles (Michel et al., 2019); at the representation level, learned embeddings concentrate semantic or functional information within specific subspaces, forming groups of coordinates that jointly encode meaningful behavior (Clark et al., 2019). Beyond model internals, LLM behavior is evaluated on groups of prompts or tasks, such as overall accuracy on MMLU (Hendrycks et al., 2021) or GSM8K (Cobbe et al., 2021).

In practice, only a small subset of groups is typically relevant, leading to a group-sparse structure and motivating group-wise variable selection (Yuan & Lin, 2006). To ensure reliability, it is crucial to control false discoveries at the group level, a problem addressed by the group knockoff filter by Dai et al. (2016). However, in many modern settings, sparsity is not specified on the original groups themselves, but instead arises through structured transformations of the underlying parameters. For example, in LLM behavior auditing, the goal is no longer to evaluate overall accuracy on a given class of questions, but to identify models that exhibit anomalous behavior compared to common models. Such behavioral anomalies are characterized by performance deviations from others over groups of inputs, rather than by raw representations or aggregate performance metrics. Beyond LLM auditing, similar forms of transformed group sparsity also arise in neuroscience and crowdsourcing analysis (Xu et al., 2018).

To fill this gap, we propose *Split Group Knockoff (SGK)*, a new framework for group-wise variable selection under grouped transformational sparsity. Our method extends the *Split Knockoff* procedure (Cao et al., 2024), which provides element-wise false discovery control for transformational sparsity, to settings where relevance is defined over groups of transformed variables. An overview of the workflow on LLMs auditing task is illustrated in Figure 1. The goal of this task is to compare the performance of multiple test LLMs against a reference model over a shared set of questions and identify the outlier models with abnormal behavior. For each model, task-specific questions and answers embeddings are treated as a model-level group and provided as input to the SGK procedure. Paired significance statistics are constructed from the original groups and their knockoff counterparts, and combined to form SGK statistics through a contrast operation. Higher values of the resulting SGK statistics indicate a greater likelihood that the corresponding

---

[1]Department of Applied Mathematics and Computational Science, University of Pennsylvania, Philadelphia, Pennsylvania, United States [2]Institute of Science and Technology for Brain-Inspired Intelligence, Fudan University, Shanghai, China [3]Shanghai Innovation Institute, Shanghai, China [4]School of Data Science, Fudan University, Shanghai, China. Correspondence to: Yanwei Fu <yanweifu@fudan.edu.cn>, Xinwei Sun <sunxinwei@fudan.edu.cn>.

*Proceedings of the 43rd International Conference on Machine Learning*, Seoul, South Korea. PMLR 306, 2026. Copyright 2026 by the author(s).

model exhibits abnormal behavioral deviation to the reference model. Models whose statistics exceed a data-driven threshold are selected as outliers, while controlling the false discovery rate (FDR) at the group level. The resulting framework accommodates a broad class of linear transformations and grouping structures, enabling principled discovery in complex applications such as multi-task comparisons and scientific analysis.

Our main contributions are summarized as follows:

- We address a previously unaddressed variable selection problem in which sparsity arises at the group level in a linearly transformed parameter space with FDR control.

- We address the theoretical challenges induced by group-wise penalties in transformed spaces, and establish rigorous guarantees for group-level FDR control and support recovery consistency.

- We demonstrate the broad applicability of the proposed framework through real data applications in behavioral auditing of LLMs and brain region and connectivity selection for Alzheimer's Disease.

## 2. Related Work

A common approach to modeling sparse data is to estimate regression coefficients by optimizing a loss function augmented with a regularization term. One of the most influential methods in this class is the LASSO (Tibshirani, 1996), which induces sparsity through an $\ell_1$ penalty and has become a cornerstone of modern high-dimensional statistics. Building on this idea, Group LASSO was proposed to address scenarios where predictors exhibit a known group structure, enabling selection at the group level rather than at the level of individual variables (Yuan & Lin, 2006). To further extend sparse modeling beyond unstructured sparsity, the Generalized LASSO was introduced to capture structured sparsity patterns by encouraging sparsity in linear transformations of the coefficients (Tibshirani, 2011).

Beyond estimation, high-dimensional inference requires controlling false discoveries from multiple testing, with the false discovery rate (FDR) emerging as a central error criterion (Benjamini & Hochberg, 1995). Since its introduction, FDR has motivated extensive methodological development, including the classical Benjamini–Hochberg procedure based on ordered p-values (Benjamini & Hochberg, 1995). More recently, the knockoff framework has become a central approach for FDR control in variable selection, offering greater flexibility under general and non-orthogonal designs (Barber & Candès, 2015). The knockoff methodology has since been extended to grouped variables, random design settings, nonparametric models, and high-dimensional

regimes where the number of variables exceeds the sample size (Dai & Barber, 2016; Candes et al., 2018; Barber & Candès, 2019). Notably, recent work has explored sparsity in linear transformations of regression coefficients, yet existing methods do not address settings where such transformed parameters exhibit group sparsity (Cao et al., 2024). This gap motivates the development of new knockoff procedures with rigorous FDR control under more structured sparsity assumptions.

## 3. Split Group Knockoff

We consider the linear model

$$y = X\beta^* + \varepsilon, \qquad \gamma^* = D\beta^*, \qquad (1)$$

where $y \in \mathbb{R}^n$ is the response vector, $X \in \mathbb{R}^{n \times p}$ is the design matrix, $\varepsilon \sim \mathcal{N}(0, \sigma^2 I_n)$ is Gaussian noise, and $\beta^*$ is the coefficient vector. We introduce $\gamma^* \in \mathbb{R}^m$ as the transformed vector, defined by a pre-defined transformation matrix $D \in \mathbb{R}^{m \times p}$.

We assume that $\gamma^*$ exhibits a sparse group structure. Specifically, its entries are partitioned into $k$ disjoint groups,

$$\mathbb{G}_1 \cup \mathbb{G}_2 \cup \cdots \cup \mathbb{G}_k = \{1, 2, \ldots, m\},$$

where $\mathbb{G}_l$ denotes the index set of the $l$-th group. Let $\gamma_l \in \mathbb{R}^{g_l}$ denote the subvector of the $l$-th group, with group size $g_l \stackrel{\text{def}}{=} |\mathbb{G}_l|$. We assume that only a small fraction of these groups contains nonzero signals. Accordingly, define the true null and non-null group sets as

$$S_0 \stackrel{\text{def}}{=} \{l : \gamma_l^* = 0\}, \qquad S_1 \stackrel{\text{def}}{=} \{l : \gamma_l^* \neq 0\},$$

and let $\widehat{S} \subset \{1, 2, \ldots, k\}$ denote an estimated support set. Our goal is to develop a procedure for recovering $\widehat{S}$ while controlling the group-wise false discovery rate (FDR) at a target nominal level $q$. The group-wise FDR is defined as

$$\text{FDR}_{\text{group}} = \mathbb{E}\left[ \frac{\left| \left\{ l : l \in \widehat{S} \cap S_0 \right\} \right|}{\left| \widehat{S} \right| \vee 1} \right].$$

In the task of identifying outliers among $R$ large language models, we assume that each model $r \in [R]$ admits an associated coefficient vector $\beta^{r,*} \in \mathbb{R}^{\widetilde{p}}$ with $\widetilde{p} = p/R$, which encodes its predictive behavior mapping questions $X$ to responses $y$. Without loss of generality, we assume the first model as the reference inlier model. To identify the outlier model, the transformation matrix $D \in \mathbb{R}^{pR \times p}$ is defined by $\gamma^{r-1,*} \stackrel{\text{def}}{=} (D\beta^*)(r, 1) = \beta^{r,*} - \beta^{1,*}$. That means, $\gamma^{r-1,*}$ for $r \geq 2$ quantifies the deviation of the $r$-th model's behavior from that of the reference model.

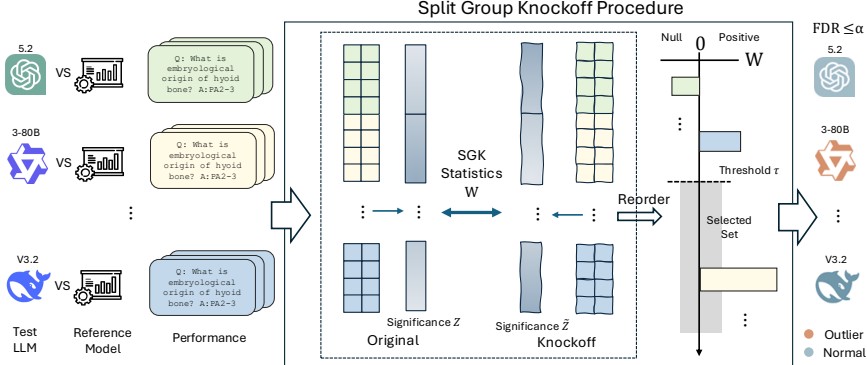

*Figure 1.* Overview of the proposed Split Group Knockoff (SGK) framework for LLM behavior auditing. For each subject domain, model-specific question–response embeddings are compared against a reference model to construct behavioral deviation statistics. SGK then computes group importance statistics $W$ using original and knockoff variables, and identifies models with significant behavioral deviations under FDR control. Orange models denote detected outliers and blue models denote behaviorally consistent models.

## 3.1. Procedure of Split Group Knockoff

Our procedure is built upon the following *Split Group Lasso* model for jointly estimating $\beta^*$ and $\gamma^*$. In particular, we introduce the variable splitting strategy that relaxes the hard constraint $\gamma = D\beta$ by allowing a quadratic deviation. This leads to the following group-Lasso optimization objective

$$\min_{\beta,\gamma} \frac{1}{2n}\|y - X\beta\|_2^2 + \frac{1}{2\nu}\|D\beta - \gamma\|_2^2 + \lambda\|\gamma\|_{\text{group}}. \quad (2)$$

The objective can be viewed as the extension of *Split Lasso* (Cao et al., 2024) from $\ell_1$ penalty to group penalty $\|\gamma\|_{\text{group}} = \sum_{l=1}^{k} \|\gamma_l\|_2$. In the variable splitting term, $\nu > 0$ controls the proximity between $D\beta$ and $\gamma$. As $\nu \to 0$, the penalty becomes $\|D\beta\|_{\text{group}}$. However, a nonzero $\nu$ turns to penalize $\gamma$ with an identity matrix. As will be shown later, this orthogonal structure brings benefits in constructing the Knockoff copies and model selection consistency.

To construct valid knockoffs, we randomly split the dataset $\mathcal{D} = (X, y)$ into two independent subsets $\mathcal{D}_1 = (X_1, y_1)$ and $\mathcal{D}_2 = (X_2, y_2)$ with sample sizes $n_1$ and $n_2$, respectively, where $n_1 + n_2 = n$ and $n_2 \geq m + p$. We will first discuss the situation where $n_2 < m + p$ in Section 3.2. Our procedure is composed of two steps, with the first step estimating $\beta$ on $\mathcal{D}_1$, and the second step constructing the selection set $\widehat{S}$ with FDR control guarantee.

*Step I. Estimating $\widehat{\beta}(\lambda)$ on $\mathcal{D}_1$.* We either employ (2) to estimate the solution path $\beta(\lambda)$, or use the cross-validation estimator $\widehat{\beta}_{\widehat{\lambda},\widehat{\nu}}$, with hyperparameters $\widehat{\lambda}, \widehat{\nu}$ chosen via cross-validation.

*Step II. Constructing $\widehat{S}$ via Knockoffs on $\mathcal{D}_2$.* First, we write (1) in a lifted model

$$\widetilde{y} = A_\beta \beta^* + A_\gamma \gamma^* + \widetilde{\varepsilon},$$

where the response $\widetilde{y}$, design $A_\beta, A_\gamma$, and the noise $\varepsilon$ are

$$\widetilde{y} = \begin{pmatrix} y_2/\sqrt{n_2} \\ 0_m \end{pmatrix}, \qquad A_\beta = \begin{pmatrix} X_2/\sqrt{n_2} \\ D/\sqrt{\nu} \end{pmatrix},$$

$$A_\gamma = \begin{pmatrix} 0_{n_2 \times m} \\ -I_m/\sqrt{\nu} \end{pmatrix}, \qquad \widetilde{\varepsilon} = \begin{pmatrix} \varepsilon_2/\sqrt{n_2} \\ 0_m \end{pmatrix}. \quad (3)$$

We then construct a Knockoff copy $\widetilde{A}_\gamma$ by mimicking the covariance behavior of $A_\beta$ and $A_\gamma$,

$$A_\beta^\top \widetilde{A}_\gamma = A_\beta^\top A_\gamma, \quad \widetilde{A}_\gamma^\top \widetilde{A}_\gamma = A_\gamma^\top A_\gamma$$
$$A_\gamma^\top \widetilde{A}_\gamma = A_\gamma^\top A_\gamma - \text{diag}(s), \quad (4)$$

where $s \in \mathbb{R}_+^m$ is a nonnegative vector that is selected to be as large as possible to enhance the power of the procedure[1]. We next compute the importance score $Z_l$ for $A_{\gamma,l}$ and $\widetilde{Z}_l$ for $\widetilde{A}_{\gamma,l}$. With estimated $\widehat{\beta}(\lambda)$, we optimize the solution paths of $\{\gamma(\lambda), \widetilde{\gamma}(\lambda)\}_\lambda$ via the following

$$\gamma(\lambda) = \arg\min_\gamma \frac{1}{2}\|\widetilde{y} - A_\beta\beta(\lambda) - A_\gamma\gamma\|_2^2 + \lambda\|\gamma\|_{\text{group}},$$
$$\widehat{\gamma}(\lambda) = \arg\min_{\widetilde{\gamma}} \frac{1}{2}\|\widetilde{y} - A_\beta\beta(\lambda) - \widetilde{A}_\gamma\widetilde{\gamma}\|_2^2 + \lambda\|\widetilde{\gamma}\|_{\text{group}}. \quad (5)$$

We then define $Z_l$ (*i.e.*, $\widetilde{Z}_l$) as the emergence time along the solution path, at which $\gamma_l$ (*i.e.*, $\widetilde{\gamma}_l$) is selected

$$Z_l \overset{\text{def}}{=} \sup\{\lambda : \gamma_l(\lambda) \neq 0\}, \quad \widetilde{Z}_l \overset{\text{def}}{=} \sup\{\lambda : \widetilde{\gamma}_l(\lambda) \neq 0\}.$$

The hyperparameter $\lambda$ determines the strength of penalization. Therefore, the group feature that is selected at a larger

---

[1]Details of computing $\widetilde{A}_\gamma$ optimizing $s$ can be found in Appendix A.

value of $\lambda$ is considered more important. With $Z_l$ and $\widetilde{Z}_l$, we then construct the comparison statistics as follows,

$$W_l \overset{\text{def}}{=} Z_l \cdot \text{sign}(Z_l - \widetilde{Z}_l), \qquad l = 1, \ldots, k.$$

An important non-null group feature is expected to exhibit a large positive value of $W$, whereas for a null feature, the importance of the original feature is comparable to, or even smaller than, that of its knockoff copy. Consequently, groups with larger positive values of $W_l$ are more likely to correspond to true signals and should be selected preferentially. Similar to the standard Knockoff procedure (Barber & Candès, 2015), we define a data-adaptive threshold for feature selection. Specifically, given a target FDR level $q \in (0, 1)$, the threshold $T_q^a$ is defined as

$$T_q^a = \min \left\{ \lambda : \frac{a + |\{l : W_l \leq -\lambda\}|}{1 \vee |\{l : W_l \geq \lambda\}|} \leq q \right\}, \quad (6)$$

or $= +\infty$ if the set is empty. Here, $T_q^0$ for $a = 0$ stands for the Split Group Knockoff (SGK) procedure, while $T_q^1$ represents the procedure of Split Group Knockoff+. Accordingly, SGK and SGK+ select the non-null groups by $\widehat{S} \overset{\text{def}}{=} \{l : W_l \geq T_q^a\}$ with $a = 0, 1$, respectively.

We next provide a theoretical guarantee for the proposed procedure. The following theorem establishes control of the modified group-FDR at level $q$.

**Theorem 3.1** (FDR Control of Split Group Knockoffs). *Given a nominal level $q > 0$, the following holds all $\nu > 0$:*

*(i) (**Modified group-FDR of Split Group Knockoff**).*

$$\mathbb{E}\left[ \frac{\left|\left\{l : l \in \widehat{S} \cap S_0\right\}\right|}{\left|\widehat{S}\right| + q^{-1}} \right] \leq q.$$

*(ii) (**group-FDR of Split Group Knockoff+**).*

$$\mathbb{E}\left[ \frac{\left|\left\{l : l \in \widehat{S} \cap S_0\right\}\right|}{\left|\widehat{S}\right| \vee 1} \right] \leq q.$$

The difference between the modified FDR and the standard FDR in Theorem 3.1 becomes negligible when the number of selected groups $|\widehat{S}|$ is large, in which case the two methods are effectively equivalent. The proof of Thoerem 3.1 is provided in Appendix B.1.

*Remark* 3.2 (Extension to multivariate responses). The proposed procedure is presented under the setting of a univariate response $y \in \mathbb{R}^n$, where $\beta$ and $\gamma$ are vector-valued parameters. It can be trivially extended to multivariate response models with $Y \in \mathbb{R}^{n \times q}$. In this scenario, $\beta$ and $\gamma$ become matrix-valued, and all $\ell_2$ norms appearing in the

optimization objectives are replaced by the corresponding Frobenius norms. With this modification, all other parts of the procedure can be applied without change, including the construction of the split knockoff statistics and the resulting group-FDR control. More details can be found in Appendix C.

## 3.2. High-Dimensional Extension

We now consider the high-dimensional regime in which $n_2 \geq m + p$ is not satisfied. Without this condition, we cannot construct $\widetilde{A}_\gamma$ that satisfies the covariance constraint (4). To address this issue, we introduce a screening step on $\mathcal{D}_1$ for dimension reduction before applying the knockoff procedure on $\mathcal{D}_2$.

Specifically, we use $\mathcal{D}_1 = (X_1, y_1)$ to conduct preliminary screening and obtain estimated support sets $\widehat{S}_\beta$ and $\widehat{S}_\gamma$ for $\beta$ and $\gamma$, until $n_2 \geq m + p$ is satisfied. After that, we apply the Split Group Knockoff procedure in the above section to $\widehat{S}_\beta$ and $\widehat{S}_\gamma$. For the feature screening, we can apply Group Lasso (Yuan & Lin, 2006) or Group Sure Independence Screening (SIS) (Niu et al., 2020), both of which are guaranteed to retain all non-null features under suitable conditions.

Let $X_{\widehat{S}_\beta}$ denote the submatrix of $X_2$ consisting of columns indexed by $\widehat{S}_\beta$, and let $D_{\widehat{S}_\beta, \widehat{S}_\gamma}$ denote the submatrix of $D$ with columns indexed by $\widehat{S}_\beta$ and rows indexed by $\widehat{S}_\gamma$. Using these screened quantities, we define the augmented response vector and design matrices as follows:

$$\widetilde{y} = \begin{pmatrix} y_2/\sqrt{n_2} \\ 0_{|\widehat{S}_\gamma|} \end{pmatrix}, \qquad A_\beta = \begin{pmatrix} X_{\widehat{S}_\beta}/\sqrt{n_2} \\ D_{\widehat{S}_\beta, \widehat{S}_\gamma}/\sqrt{\nu} \end{pmatrix},$$

$$A_\gamma = \begin{pmatrix} 0_{n_2 \times |\widehat{S}_\gamma|} \\ -I_{|\widehat{S}_\gamma|}/\sqrt{\nu} \end{pmatrix}, \qquad \widetilde{\varepsilon} = \begin{pmatrix} \varepsilon_2/\sqrt{n_2} \\ 0_{|\widehat{S}_\gamma|} \end{pmatrix}. \tag{7}$$

With some slight abuse of notation, we reuse the symbols $\widetilde{y}$, $A_\beta$, $A_\gamma$, and $\widetilde{\varepsilon}$ as defined in (3). We then apply the remaining steps of the procedure from the previous section to construct the selection set $\widehat{S}$. The following theorem guarantees FDR control, assuming that the estimated supports $\widehat{S}_\beta$ contains the true supports of $\beta^*$ and $\gamma^*$, respectively.

**Theorem 3.3** (Conditional FDR Control in High Dimensions). *Let $S_\beta$ denote the true support of $\beta$, and let*

$$\Upsilon \overset{\text{def}}{=} \{S_\beta \subseteq \widehat{S}_\beta\}.$$

*Then, for all $0 < q \leq 1$ and all $\nu > 0$, the following holds:*

*(i) (Modified group-FDR of Split Group Knockoff).*

$$\mathbb{E}\left[ \frac{|\{l : l \in \widehat{S} \cap S_0\}|}{|\widehat{S}| + q^{-1}} \,\middle|\, \Upsilon \right] \leq q.$$

*(ii) (group-FDR of Split Group Knockoff+).*

$$\mathbb{E}\left[\left.\frac{|\{l : l \in \widehat{S} \cap S_0\}|}{|\widehat{S}| \vee 1}\right| \Upsilon\right] \leq q.$$

The proof of Theorem 3.3 is in Apenndix B.2.

### 3.3. Model Selection Consistency

In this section, we investigate the model selection consistency of the Split Group Lasso to elucidate the selection power of our procedure. Specifically, let $\widehat{S}(\lambda) \stackrel{\text{def}}{=} \{l \leq k : \widehat{\gamma}_l(\lambda) \neq 0\}$ denote the set of selected groups. We show that, for an appropriate choice of $\lambda$, $\widehat{S}(\lambda)$ can asymptotically recover the true support set $S_1$.

We begin by introducing the conditions required for our analysis. Define

$$H_\nu \stackrel{\text{def}}{=} I_m - \frac{1}{\nu} D(\Sigma_X + L_D)^{-1} D^\top,$$

where $\Sigma_X = X^\top X / n$ and $L_D = D^\top D / m$. Let $H_\nu^{11}$ and $H_\nu^{00}$ denote the submatrices of $H_\nu$ corresponding to the true support set $S_1$ and its complement $S_0$, respectively. Similarly, let $H_\nu^{10}$ and $H_\nu^{01}$ denote the cross-covariance submatrices. For each group $G_l$ with $l \in S_0$, we denote by $H_\nu^{01,l}$ the submatrix of $H_\nu^{01}$ associated with group $G_l$.

**Assumption 3.4.** The design matrix $X$ and the transformation matrix $D$ satisfy the following conditions:

1. *Restricted strong convexity.* The minimum eigenvalue of $H_\nu^{11}$ is bounded below by a constant $C_{\min} > 0$.

2. *Bounded design.* The columns of $X$ are normalized such that $\max_{i \in [p]} \|x_i\|_2 / \sqrt{n} \leq 1$, and there exists a constant $\kappa > 0$ satisfying

$$\left\|D(\Sigma_X + L_D)^{-1} \frac{X^\top}{\sqrt{n}}\right\|_2 \leq \kappa.$$

3. *$\nu$-incoherence.* There exists a parameter $\chi_\nu \in (0, 1]$ such that

$$\max_{l \in S_0}\left\|H_\nu^{01,l}(H_\nu^{11})^{-1}\right\|_2 \leq \frac{1 - \chi_\nu}{\sqrt{|S_1|}}.$$

Under this condition, we establish the no-false-positive and model selection consistency of Split Group Lasso.

**Proposition 3.5** (Model Selection Consistency). *Under Assumption 3.4, there exists a constant $C > 0$ such that for any regularization sequence $\{\lambda_n\}$ satisfying*

$$\lambda_n \geq \frac{4(\sqrt{|S_1|} + 1 - \chi_\nu)C\sigma}{\sqrt{|S_1|}\nu\chi_\nu}\sqrt{\frac{m}{n}},$$

*following properties hold with probability at least $1 - e^{-m}$:*

*(i) (No false positives). The Group Split Lasso problem* (2) *admits a unique solution $(\widehat{\beta}, \widehat{\gamma})$. Additionally, no false positive groups are selected, that is, $\text{supp}(\widehat{\gamma}) \subseteq S_1$.*

*(ii) (Sign-consistency) Under the following $\gamma_{\min}$ condition, $\widehat{\gamma}$ recovers the true support set of $\gamma^*$, that is, $\widehat{S}(\lambda_n) = \{l : \widehat{\gamma}_l(\lambda_n) \neq 0\} = S_1$,*

$$\min_{i \in S_1} \gamma_i^* > \frac{\nu\sigma}{C_{\min}}\sqrt{\frac{2m}{n}} + \lambda_n \nu \left\|(H_\nu^{11})^{-1}\right\|_\infty.$$

The proof is provided in Appendix B.3. The hyperparameter $\nu$ governs a trade-off between the incoherence condition and selection power. As $\nu \to \infty$, $H_\nu = I_m - \frac{1}{\nu}D[\Sigma_X + L_D]^{-1}D^T \to I_m$, so $H_\nu^{01} \to \mathbf{0}_{|S_0|\times|S_1|}$ and $H_\nu^{11} \to I_{|S_1|}$. Therefore, $\max_{l \in S_0}\left\|H_\nu^{01,l}(H_\nu^{11})^{-1}\right\|_2 \to 0$, which guarantees the absence of false positives along the early stages of the selection path, in the sense that all variables selected initially must belong to the true signal set. However, excessively large values of $\nu$ may overly regularize the model, leading to the exclusion of weak but nonnull signals. As demonstrated in the simulation studies, an appropriately chosen $\nu$ strikes a balance between these competing effects and yields optimal selection power.

## 4. Numerical Experiments

We evaluate the finite-sample performance of the proposed Split Group Knockoff procedure on synthetic data. We repeat each experiment 10 times and report the empirical FDR and power. We first study the baseline block-structured signal setting under two choices of the linear transformation matrix $D$: the identity matrix and the one-dimensional fused Lasso operator. We then further examine the robustness of the fused Lasso setting by varying the correlation level $c$ and the signal-to-noise ratio, controlled through the signal magnitude $A$ and noise standard deviation $\sigma$.

*Data generation.* Responses are generated according to (1), where the sample size is $n = 1000$ and the dimension is $p = 200$. The rows of the design matrix $X \in \mathbb{R}^{n \times p}$ are drawn independently from $\mathcal{N}(0, \Sigma)$, with $\Sigma_{ii} = 1$ and $\Sigma_{ij} = c^{|i-j|}$. Here $c$ controls the correlation strength among nearby features; larger values of $c$ correspond to stronger feature dependence. In the baseline experiment, we set $c = 0.5$. Once generated, $X$ is fixed across all trials.

Let $0 = s_0 < s_1 < \cdots < s_K = p$ denote the boundary indices of a partition of $\{1, \ldots, p\}$ into $K$ contiguous blocks, where the first block has size $s_1 - s_0 = 10$ and the remaining blocks satisfy $s_k - s_{k-1} = 5$ for $k \geq 2$. The true coefficient vector $\beta^*$ is specified blockwise as

$$\beta_j^* = \begin{cases} A\dfrac{j - s_{k-1}}{s_k - s_{k-1}}, & j \in (s_{k-1}, s_k], \ k \bmod 2 = 1, \\ 0, & j \in (s_{k-1}, s_k], \ k \bmod 2 = 0. \end{cases}$$

Here $A$ denotes the signal magnitude. The response vector is generated as
$$y = X\beta^* + \varepsilon,$$
where $\varepsilon \sim \mathcal{N}(0, \sigma^2 I_n)$ and $\sigma$ denotes the noise standard deviation. In the baseline experiment, we set $A = 10$ and $\sigma = 1$.

*Identity transformation.* When $D$ is set to the identity matrix $I_p$, we have $\gamma^* = \beta^*$. The group structure coincides with the block partition of $\beta^*$, and a group is non-null if and only if its index $k$ is odd.

*Fused Lasso transformation.* When $D \in \mathbb{R}^{(p-1) \times p}$ is set to be the one-dimensional fused Lasso operator, we have $\gamma^* = D\beta^*$. The vector $\gamma^*$ is naturally divided into $K = 59$ contiguous groups with group sizes
$$(g_1, \ldots, g_{59}) = (10, 4, 6, 4, 6, \ldots, 4, 6, 4, 5),$$
where the last group has size 5 so that $\sum_{l=1}^{59} g_l = m = p - 1$. Odd-indexed groups are nonzero with a constant negative signal and a single positive spike at the end, while even-indexed groups are identically zero; the last group contains only the negative signal.

*Additional robustness settings.* To assess the effect of feature dependence, we keep $A = 10$ and $\sigma = 1$ fixed and vary the correlation parameter over $c \in \{0.5, 0.8, 0.9\}$ under the fused Lasso transformation. To assess the effect of signal strength and noise level, we keep $c = 0.5$ fixed and compare
$$(A, \sigma) \in \{(10, 1), (7, 1.5), (5, 2)\}.$$

The pair $(A, \sigma)$ determines the effective signal-to-noise ratio: smaller $A$ and larger $\sigma$ make the non-null groups harder to detect.

*Implementation details.* We consider two choices to compute $\beta(\lambda)$. First, we employ the `glmnet` package (Friedman et al., 2010) to compute the solution path $\beta(\lambda)$ via (2) along the $\nu$-Split Lasso solution path computed on $\mathcal{D}_1$. When $D = I_p$, we take $\log(\nu)$ ranging from 0 to 2 with a step size 0.2, whereas when $D = D_G$, we take $\log(\nu)$ ranging from 1 to 3 with a step size 0.2. For the second choice, we take $\beta(\lambda)$ as the cross-validated optimum on $\mathcal{D}_1$, with the optimal $\widehat{\lambda}$ chosen from 1 to $10^{-8}$ with a step size 0.4, and $\log(\widehat{\nu})$ selected over the same range as in the path-based procedure. To compute the $Z_l$ and $\widetilde{Z}_l$ for each group, we employ `glmnet` to compute the solution path with $\lambda$ ranging from 1 to $10^{-6}$, with a step size 0.01.

*Results.* Figure 2 summarizes the baseline FDR and power curves with respect to $\log \nu$. The proposed procedure consistently controls FDR under the nominal level while maintaining high power with small variance. As $\nu$ increases, FDR control becomes more conservative, a pattern also reported in non-groupwise sparsity settings (Cao et al., 2024).

Furthermore, the cross-validated estimator achieves greater power compared to the solution-path-based estimator. Under the solution-path implementation, the selection power initially increases with $\nu$ for small values, but subsequently decreases as $\nu$ becomes large. This phenomenon reflects the trade-off induced by $\nu$: larger values strengthen the incoherence condition and suppress false positives, while simultaneously reducing sensitivity to weak but non-null features.

Figure 3 reports the additional fused-Lasso experiments. Across the correlation sweep, the procedure remains stable as $c$ increases from 0.5 to 0.9, although stronger correlation makes the selection problem more challenging. Across the signal-to-noise sweep, decreasing $A$ and increasing $\sigma$ leads to lower power, as expected, while the FDR remains controlled. These results suggest that the proposed Split Group Knockoff procedure is robust to both stronger feature dependence and weaker signal regimes.

## 5. Behavior Auditing of LLM

### 5.1. Modeling and Experimental Setup

To study how large language models exhibit heterogeneous behaviors across subject domains, we perform task-level model auditing beyond aggregate benchmark scores by applying our method to the MMLU-Pro benchmark (Wang et al., 2024), a rigorous testbed comprising approximately 12k complex multiple-choice questions across 14 diverse subject domains. To enhance sensitivity, we retain questions with high cross-model heterogeneity using an entropy-based threshold, resulting in $T = 13$ tasks with sample sizes $s_t \in [96, 616]$, $t \in \{1, 2, \ldots, T\}$ (see Appendix D).

We evaluate $M = 6$ state-of-the-art LLMs, including DeepSeek-V3.2 (DeepSeek-AI, 2025), DeepSeek-V3.2-Speciale (DeepSeek-AI, 2025), Qwen3-Next-80B-A3B-Thinking (Qwen Team, 2025; Yang et al., 2025), Qwen3-Next-80B-A3B-Instruct (Qwen Team, 2025; Yang et al., 2025), Qwen3-235B-A22B (Qwen Team, 2025), and GPT-5.2 (OpenAI, 2025).

Each question and model-selected option is embedded using nomic-embed-text-v2-moe (Nussbaum & Duderstadt, 2025). We use the first $d = 64$ dimensions of the Matryoshka embedding and apply min–max normalization to scale values to $[0, 1]$. For model $a$ and task $t$ with $q_t$ questions, the question and response embeddings form matrices $X_a^t, Y_a^t \in \mathbb{R}^{q_t \times d}$.

Then, the task-specific design matrix $X^t \in \mathbb{R}^{\sum M q_t \times Md}$ is constructed by arranging all $X_a^t$ along the block diagonal, so that each block corresponds to one model and shares the same question representations. Stacking all response embeddings $Y_a^t$ yields the response matrix $Y \in \mathbb{R}^{M q_t \times d}$.

We model the relationship between question and response

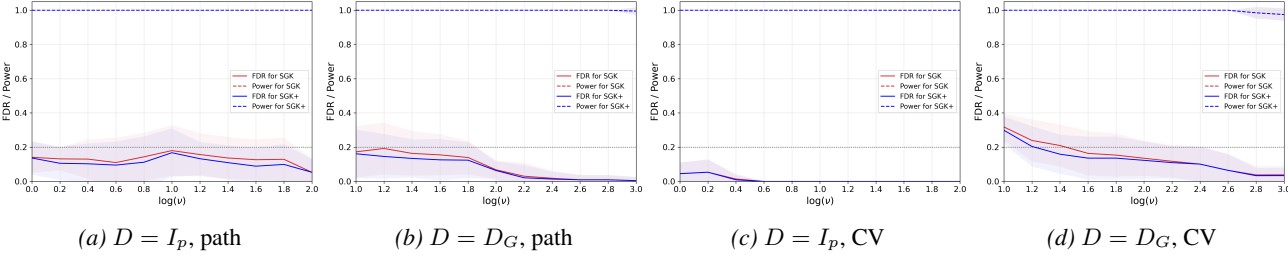

*(a) $D = I_p$, path*      *(b) $D = D_G$, path*      *(c) $D = I_p$, CV*      *(d) $D = D_G$, CV*

*Figure 2.* Empirical FDR and power under the baseline block-structured signal setting. Results are shown for the identity transformation ($D = I_p$) and the fused Lasso transformation ($D = D_G$), using both solution-path-based and cross-validated estimators.

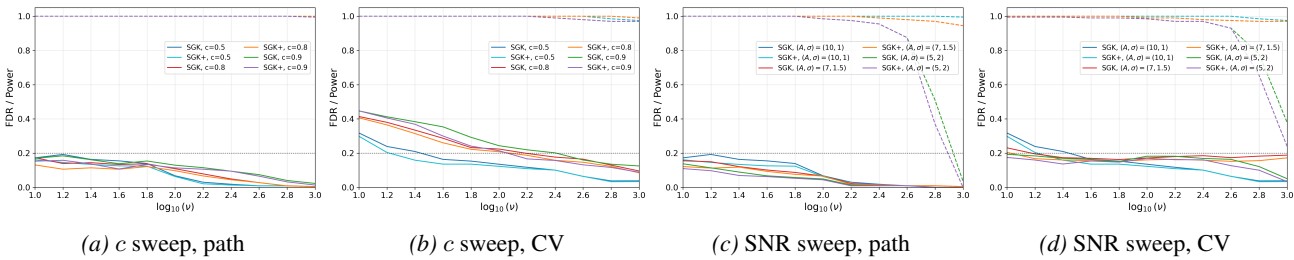

*(a) $c$ sweep, path*      *(b) $c$ sweep, CV*      *(c) SNR sweep, path*      *(d) SNR sweep, CV*

*Figure 3.* Empirical FDR and power under additional fused Lasso settings ($D = D_G$). The first two panels vary the feature correlation parameter $c$ while fixing $(A, \sigma) = (10, 1)$. The last two panels vary the signal magnitude and noise level through $(A, \sigma) \in \{(10, 1), (7, 1.5), (5, 2)\}$ while fixing $c = 0.5$. Both solution-path-based and cross-validated estimators are reported.

embeddings through the multivariate linear model $Y = X^t B + E$, where $B \in \mathbb{R}^{Md \times d}$ is the coefficient matrix characterizing model-specific behaviors. Due to the block-diagonal structure of $X^t$, the coefficient matrix $B$ admits a corresponding blockwise structure, where each submatrix represents the behavior of model $a$ under topic $t$.

To characterize model-specific behavioral deviations for a fixed task, we introduce a contrast operator relative to a reference model indexed by $K$. We define $D = \widetilde{D} \otimes I_d \in \mathbb{R}^{(M-1)d \times Md}$, where $\widetilde{D} \in \mathbb{R}^{(M-1) \times M}$ is a contrast matrix whose $i$-th row contains a 1 in the column corresponding to model $i \neq K$, a $-1$ in the column corresponding to model $K$, and zeros elsewhere, and $I_d$ denotes the $d \times d$ identity matrix.

Applying $D$ to the blockwise behavior matrix $B$ yields the deviation matrix $\Gamma = DB \in \mathbb{R}^{(M-1)d \times d}$, which stacks the deviations of each model from the reference. Due to the block structure of $B$, the deviation matrix $\Gamma$ admits a corresponding blockwise decomposition $\Gamma = \left[ (B_1 - B_K)^\top, \cdots, (B_{K-1} - B_K)^\top, (B_{K+1} - B_K)^\top, \cdots, (B_M - B_K)^\top \right]^\top$, where each block $B_a - B_K \in \mathbb{R}^{d \times d}$ represents the task-specific behavioral deviation of model $a$ relative to the reference. We treat each block of $\Gamma$ as a group, and sparsity at the group level indicates that only a small subset of models exhibit abnormal behavior under the given task. To establish a robust baseline, we apply the ARMUL framework (Duan & Wang, 2023) to estimate the latent consensus behavior of the model ensemble, and choose the model closest to this consensus as the reference.

Since the response in our setting is matrix-valued, we adopt the multivariate-response version of the split group knockoff procedure described in Section C. For each task $t$, we randomly split the data into two disjoint subsets $\mathcal{D}_1$ and $\mathcal{D}_2$ with a ratio $|\mathcal{D}_1|/|\mathcal{D}_2| = 0.35$. We estimate the behavior parameter $B$ on $\mathcal{D}_1$ using ridge regression to obtain $\widehat{B}$, and apply the split group knockoff method on $\mathcal{D}_2$ to identify deviations $\Gamma$. The regularization parameters are selected over a grid with $\lambda \in 10^{\{0, -0.01, \ldots, -6\}}$ and $\nu = 10^3$. Throughout all experiments, the target FDR is set to $q = 0.2$.

### 5.2. Results

We begin with an overview of behavior auditing results across all subject domains. Table 1 summarizes the outlier models identified with the proposed split group knockoff procedure for each subject. Overall, SGK flags a small number of models in only a subset of subjects, suggesting that model behavior is largely consistent, with deviations concentrated in specific domains. To disentangle behavioral deviation from raw performance, we additionally report subject-level entropy in Table 1 and model accuracy in Table 4 (Appendix E).

These findings are supported by the descriptive statistics of the underlying data. From the perspective of model accuracy, subjects in which SGK identifies outlier models often exhibit substantial performance disparities across models. For example, in *Psychology*, the model flagged as abnormal attains an accuracy of approximately $41\%$, which is markedly lower than that of the remaining models, whose

*Table 1.* Outlier LLMs selected by SGK. The Entropy column quantifies the heterogeneity of model performance within subject.

| Subject | Selected Model | Entropy |
|---|---|---|
| Engineering | - | 1.34 |
| Chemistry | GPT-5 | 1.036 |
| Physics | Qwen-Thk, GPT-5 | 1.034 |
| Law | - | 1.031 |
| Comp. Sci. | Qwen-Thk, Qwen-Ins, GPT-5 | 1.0247 |
| Business | - | 0.99 |
| Philosophy | - | 0.974 |
| Health | GPT-5 | 0.97 |
| Psychology | Qwen-Ins | 0.944 |
| History | - | 0.928 |
| Math | - | 0.925 |
| Biology | DS-v3, DS-Spec, Qwen-Hg | 0.891 |
| Economics | - | 0.88 |

accuracies primarily range from $48\%$ to $64\%$. A similar pattern is observed at the subject level. Subjects with higher entropy, such as *Chemistry* (1.036), *Physics* (1.034), and *Computer Science* (1.024), tend to yield one or more detected deviations, whereas low-entropy subjects including *Math* (0.925) and *Economics* (0.88) show no statistically significant differences across models. These observations suggest that the deviation patterns uncovered by SGK are closely aligned with intrinsic heterogeneity in the underlying tasks.

Beyond patterns that align with simple performance statistics, SGK also uncovers behavioral deviations that are not readily explained by accuracy or entropy alone, suggesting sensitivity to deeper structural differences in model reasoning. A notable example arises in the *Health* domain, where GPT-5 is consistently identified as exhibiting distinct behavior relative to other models, despite competitive overall accuracy. This finding aligns with recent evidence that GPT-5 employs more advanced reasoning strategies in medical decision-making, including multimodal integration and zero-shot chain-of-thought reasoning (Wang et al., 2025). The deviation detected by SGK may therefore reflect differences in reasoning style and inference structure, rather than surface-level predictive performance.

In contrast, no significant behavioral deviations are detected in subjects such as *Law*, *Business*, *Philosophy*, and *History*, where model behaviors remain highly consistent. This observation is consistent with findings reported in MMLU-Pro (Wang et al., 2024), where these domains are characterized by relatively homogeneous question sources and well-established knowledge, leading to highly consistent model performance. As a result, models exhibit similar response patterns, leaving little room for detectable behavioral

divergence.

# 6. Alzheimer's Disease

In addition to LLM behavior auditing, we also evaluate the proposed method in selecting abnormal brain regions and brain connectivity on Alzheimer's Disease Neuroimaging Initiative (ADNI)[2]. The dataset contains $n = 752$ subjects, including 126 patients with Alzheimer's disease (AD), 433 subjects with mild cognitive impairment (MCI), and 193 normal controls (NC). Brain images are parcellated using the Automatic Anatomical Labeling (AAL) atlas, which partitions the cerebrum into $p_0 = 90$ anatomical regions (Tzourio-Mazoyer et al., 2002). For each subject, the volume of each region is computed as the sum of gray matter voxels within that region.

Brain images may be acquired at different magnetic field strengths (1.5T and 3.0T). To preserve region-level interpretability, we group measurements from both field strengths, assuming that abnormal brain regions are consistent across acquisitions. We construct the design matrix $X \in \mathbb{R}^{n \times p}$ with $p = 2p_0 = 180$, where $X_{(:,2j-1)}$ and $X_{(:,2j)}$ record the regional volumes from 1.5T and 3.0T scans, respectively; missing acquisitions are set to zero. This construction naturally induces a group structure in $X$, where each pair of columns associated with the same anatomical region forms a group. All features are normalized column-wise. Besides, we set the response vector $y \in \mathbb{R}^n$ to measure Alzheimer's Disease Assessment Scale (ADAS) score (Zec et al., 1992) for all patients. In all experiments, the target FDR level is set to $q = 0.2$.

## 6.1. Brain Region Selection

For the brain region selection, we set $D = I_p$. To maximize selection power, we estimate $\widehat{\beta}_{\widehat{\lambda}, \widehat{\nu}}$ via cross-validation, where $\log \widehat{\nu}$ is chosen from $\{-0.2, 0, \dots, 2\}$ and $\log \widehat{\lambda}$ is chosen from $\{0, -0.2, \dots, -4\}$. We split the datasets into $\mathcal{D}_1$ and $\mathcal{D}_2$, with $n_1 = 150$ in $\mathcal{D}_1$ and $n_2 = 602$ in $\mathcal{D}_2$. To remove the randomness of splitting, we repeat 10 times.

The frequency of being selected among 10 trials for each region is recorded. We summarize the top 10 regions in terms of frequency in Table 2. Most of these regions have been reported in previous studies to exhibit early degeneration. In particular, the most frequently selected regions are the left middle temporal gyrus and the bilateral Hippocampus, which have been well known to be early degenerated regions in AD. Hippocampal volume loss has been widely reported as a hallmark of memory impairment and disease severity, while temporal lobe degeneration is strongly associated with

[2]https://adni.loni.usc.edu/data-samples/adni-data/

*Table 2.* Frequently selected brain regions across 10 trials.

| Region | Frequency |
|---|---|
| Temporal_Mid_L | 0.90 |
| Hippocampus_L | 0.70 |
| Hippocampus_R | 0.70 |
| Temporal_Inf_L | 0.60 |
| Parietal_Inf_R | 0.50 |
| Heschl_R | 0.40 |
| Amygdala_L | 0.30 |
| Amygdala_R | 0.30 |
| Heschl_L | 0.30 |
| Lingual_L | 0.20 |

cognitive decline and ADAS scores (Jack Jr et al., 1992; 1997; Binder et al., 2009; Vemuri & Jack Jr, 2010). Additional regions such as the inferior temporal and inferior parietal areas have also been implicated in cortical thinning and functional disruption during disease progression (Tyler et al., 2005; Schöll et al., 2016).

### 6.2. Brain Connectivity Selection

For the connection selection, we define $D$ as the graph difference operator over the anatomical adjacency graph of brain regions, since it is believed that adjacent regions have a similar degree of degeneration during disease progression. There are $m = 926 > p = 180$ connections. As in the region selection experiment, we estimate $\widehat{\beta}_{\widehat{\lambda}, \widehat{\nu}}$ via cross validation, where the optimal $\log \nu$ and $\log \lambda$ are respectively chosen from $\{0, 0.4, \ldots, 2\}$ and $\{0, -0.01, \ldots, -6\}$. We repeat the experiment 10 times, where each time we split the data into $n_1 = 200$ samples and $n_2 = 552$ samples. Table 3 lists the top 10 most frequently selected region pairs.

*Table 3.* Frequently selected brain region connections across 10 trials.

| Region 1 | Region 2 | Frequency |
|---|---|---|
| Hippocampus_R | Thalamus_R | 0.20 |
| Hippocampus_L | Lingual_L | 0.10 |
| Hippocampus_L | Precuneus_L | 0.10 |
| Hippocampus_L | Putamen_L | 0.10 |
| Hippocampus_L | Thalamus_L | 0.10 |
| Hippocampus_L | Heschl_L | 0.10 |
| Hippocampus_R | Lingual_R | 0.10 |
| Hippocampus_R | Putamen_R | 0.10 |
| Amygdala_L | Putamen_L | 0.10 |
| Temporal_Mid_R | Occipital_Inf_R | 0.10 |
| Putamen_L | Pallidum_L | 0.10 |

Most of these connections involve regions listed in Table 2. Specifically, eight pairs are associated with the bilateral Hippocampus, while two involve the Amygdala and medial temporal lobe, which have been reported as early atrophied regions in previous studies (Poulin et al., 2011; Krasuski et al., 1998). In contrast, the connection (Putamen_L, Pallidum_L) may represent a false discovery, as there is no clear evidence indicating which of these regions experiences more severe degeneration during disease progression.

## 7. Conclusion

In this work, we introduced Split Group Knockoff, a principled framework for variable selection under transformational group sparsity with false discovery rate control. By extending split knockoff ideas to grouped transformed variables, SGK addresses a gap between modern structured data settings and existing inference tools. Both theoretical guarantees and empirical studies demonstrate that SGK reliably identifies meaningful group-level signals while controlling false discoveries.

There are several directions for future research. First, it would be of interest to integrate adaptive weighting schemes, such as the Adaptive Group LASSO (Wei & Huang, 2010), with the proposed Split Group Knockoffs framework, and to investigate whether such combinations can further improve power and yield broader conditions for consistent group selection. Second, our current formulation assumes a fixed design matrix, so extending the methodology and theoretical guarantees to random design settings remains an important open problem.

## Acknowledgements

This work was supported by Young Scientists Fund of the National Natural Science Foundation of China (Grant No. KRH2305058) and the State Key Program of National Natural Science Foundation of China under Grant No. 12331009.

## Impact Statement

This paper presents work whose goal is to advance the field of Machine Learning. There are many potential societal consequences of our work, none which we feel must be specifically highlighted here.

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

## A. Construction of Split Group Knockoff Copies

In this section, we adopt the split knockoff construction in our setting to generate knockoff copies for $A_\gamma$ (Cao et al., 2024). By Proposition A.1 in (Cao et al., 2024), we know that if $n_2 \geq m + p$, for any vector $s$ in Equation 4 satisfying

$$\text{diag}(s) \succeq 0, \qquad 2C_v - \text{diag}(s) \succeq 0,$$

where

$$C_v := \Sigma_{\gamma,\gamma} - \Sigma_{\gamma,\beta}\Sigma_{\beta,\beta}^{-1}\Sigma_{\beta,\gamma}, \quad \Sigma_{\beta,\beta} := A_\beta^\top A_\beta, \quad \Sigma_{\beta,\gamma} = \Sigma_{\gamma,\beta}^\top := A_\beta^\top A_\gamma, \quad \Sigma_{\gamma,\gamma} := A_\gamma^\top A_\gamma,$$

there exists a valid Split Knockoff matrix for $n_2 \geq m + p$:

$$\tilde{A}_\gamma = A_\gamma \left( I_m - C_v^{-1}\text{diag}(s) \right) + A_\beta \Sigma_{\beta,\beta}^{-1}\Sigma_{\beta,\gamma}C_v^{-1}\text{diag}(s) + \tilde{U}K, \tag{8}$$

where $\tilde{U} \in \mathbb{R}^{(n_2+m)\times m}$ is the orthogonal complement of $[A_\beta, A_\gamma] \in \mathbb{R}^{(n_2+m)\times(m+p)}$, and $K \in \mathbb{R}^{m\times m}$ satisfies

$$K^\top K = 2\,\text{diag}(s) - \text{diag}(s)C_v^{-1}\text{diag}(s).$$

We give two typical examples statisfying the condition. For example, we can maximize the discrepancy between knockoffs and their corresponding features by solving the following SDP:

$$\text{maximize} \quad \sum_i s_i,$$

$$\text{subject to} \quad 0 \leq s_i \leq 1, \quad \frac{\text{diag}(s)}{2} \preceq C_v.$$

Additionally, we can take

$$s_i = 2\lambda_{\min}(C_v) \wedge 1, \qquad \forall i \in \{1, 2, \dots, m\},$$

where $\lambda_{\min}(C_v)$ denotes the minimum eigenvalue of $C_v$.

## B. Proof

### B.1. Proof of Theorem 3.1

*Proof of Theorem 3.1.* We begin by showing that the condition $\mathbb{E}\left[\mathcal{M}_{T_q}(W)\right] \leq 1$ is sufficient to control the modified FDR and FDR, where

$$\mathcal{M}_{T_q}(W) = \frac{\sum_{l \in S_0} \mathbf{1}\{W_l \geq T_q\}}{1 + \sum_{l \in S_0} \mathbf{1}\{W_l \leq -T_q\}}. \tag{9}$$

For the modified FDR, note that it can be transformed as

$$\mathbb{E}\left[\frac{\sum_{l \in S_0} \mathbf{1}\{W_l \geq T_q\}}{\sum_l \mathbf{1}\{W_l \geq T_q\} + q^{-1}}\right] \leq \mathbb{E}\left[\frac{1 + \sum_l \mathbf{1}\{W_l \leq -T_q\}}{\sum_l \mathbf{1}\{W_l \geq T_q\} + q^{-1}} \cdot \frac{\sum_{l \in S_0} \mathbf{1}\{W_l \geq T_q\}}{1 + \sum_{l \in S_0} \mathbf{1}\{W_l \leq -T_q\}}\right]. \tag{10}$$

By the definition of $T_q$ in Group Split Knockoff, there holds

$$\sum_l \mathbf{1}\{W_l \leq -T_q\} \leq q \sum_l \mathbf{1}\{W_l \geq T_q\}.$$

Therefore, the right-hand side of (10) is less and equal than

$$\mathbb{E}\left[\frac{1 + q\sum_l \mathbf{1}\{W_l \geq T_q\}}{\sum_l \mathbf{1}\{W_l \geq T_q\} + q^{-1}} \cdot \frac{\sum_{l \in S_0} \mathbf{1}\{W_l \geq T_q\}}{1 + \sum_{l \in S_0} \mathbf{1}\{W_l \leq -T_q\}}\right] = q\mathbb{E}\left[\mathcal{M}_{T_q}(W)\right].$$

Similarly, for the Group Split Knockoff+, we have

$$\mathbb{E}\left[\frac{\sum_{l \in S_0} \mathbf{1}\{W_l \geq T_q\}}{1 \vee \sum_l \mathbf{1}\{W_l \geq T_q\}}\right] \leq \mathbb{E}\left[\frac{1 + \sum_l \mathbf{1}\{W_l \leq -T_q\}}{1 \vee \sum_l \mathbf{1}\{W_l \geq T_q\}} \cdot \frac{\sum_{l \in S_0} \mathbf{1}\{W_l \geq T_q\}}{1 + \sum_{l \in S_0} \mathbf{1}\{W_l \leq -T_q\}}\right]$$

$$\overset{(1)}{\leq} q\mathbb{E}\left[\frac{\sum_{l \in S_0} \mathbf{1}\{W_l \geq T_q\}}{1 + \sum_{l \in S_0} \mathbf{1}\{W_l \leq -T_q\}}\right] = q\mathbb{E}\left[\mathcal{M}_{T_q}(W)\right],$$

where "(1)" follows from the definition of $T_q$ for Group Split Knockoff+.

Next, we prove $\mathbb{E}\left[\mathcal{M}_{T_q}(W)\right] \leq 1$. For shorthand notations, we rearrange the index on of $W$, such that $|W_{(1)}| \geq |W_{(2)}| \geq \cdots \geq |W_{(k^*)}|$, where $k^* \stackrel{\text{def}}{=} |S_0|$. Further denote $B_{(l)} = 1\{W_{(l)} \leq 0\}$, then there holds

$$\frac{\sum_{l \in S_0} 1\{W_l \geq T_q\}}{1 + \sum_{l \in S_0} 1\{W_l \leq -T_q\}} = \frac{1 + \sum_{l \in S_0} 1\{|W_l| \geq T_q\}}{1 + \sum_{l \in S_0} 1\{|W_l| \geq T_q, W_l \leq 0\}} - 1,$$

$$= \frac{1 + J}{1 + B_{(1)} + B_{(2)} + \cdots + B_{(J)}} - 1, \tag{11}$$

where $J = \arg\max_{l \leq k^*}\{|W_{(l)}| \geq T_q\}$, such that $W_{(1)} \geq \ldots \geq W_{(J)} \geq T_q > W_{(J+1)} \geq \ldots \geq W_{(k^*)}$. Define the filtration $\{\mathcal{F}_j\}_{j=1}^{k^*}$ in inverse time as

$$\mathcal{F}_j = \sigma\left(\left\{\sum_{l=1}^{j} B_{(l)}, B_{(j+1)}, \cdots, B_{(k^*)}\right\}\right).$$

It is easy to verify that $J$ is a stopping time with respect to the filtration $\{\mathcal{F}_j\}_{j=1}^{k^*}$ in inverse time. Applying Lemma 1 in (Barber & Candès, 2019), we have

$$\mathbb{E}\left[\frac{1 + J}{1 + B_{(1)} + B_{(2)} + \cdots + B_{(J)}}\right] \leq \rho^{-1},$$

where $\rho \stackrel{\text{def}}{=} \min_{l \leq k^*} \mathbb{P}\{B_{(l)} = 1\}$. Applying Lemma B.1, we have $\rho \geq 1/2$, which gives us

$$\mathbb{E}\left[\mathcal{M}_{T_q}(W)\right] = \mathbb{E}\left[\frac{1 + J}{1 + B_{(1)} + B_{(2)} + \cdots + B_{(J)}}\right] - 1 \leq \rho^{-1} - 1 \leq 1.$$

This completes the proof. $\square$

Theorem 3.1 depends on Lemma B.1, which is stated below.

**Lemma B.1.** *Given any determined $\beta(\lambda)$, $|W| = Z$ are determined, with $\text{sign}(W)$ determined by $\zeta$. Then $1\{W_l \leq 0\}$ are some independent Bernoulli random variables, such that for each $l \in S_0$, there holds*

$$\mathbb{P}(W_l \leq 0) \geq \frac{1}{2}.$$

We start from the KKT condition of (5). Take the subgradient equation of (5), we get

$$-A_{\gamma_l}^T\{\widetilde{y} - A_\beta \beta(\lambda) - \sum_{l=1}^{k} A_{\gamma_l}\gamma_l(\lambda)\} + \lambda\rho_l(\lambda) = 0 \tag{12}$$

$$-\widetilde{A}_{\gamma_l}^T\{\widetilde{y} - A_\beta \beta(\lambda) - \sum_{l=1}^{k} \widetilde{A}_{\gamma_l}\widetilde{\gamma}_l(\lambda)\} + \lambda\widetilde{\rho}_l(\lambda) = 0, \tag{13}$$

where $A_{\gamma_l} \in \mathbb{R}^{(n_2+m)\times g_l}$ and $\widetilde{A}_{\gamma_l} \in \mathbb{R}^{(n_2+m)\times g_l}$ are the corresponding design matrices of $\gamma_l$ and $\widetilde{\gamma}_l$ in $A_\gamma$ and $\widetilde{A}_\gamma$ respectively. Besides, $\rho_l(\lambda)$ and $\widetilde{\rho}_l(\lambda)$ are respectively defined as

$$\rho_l(\lambda) \in \partial\|\gamma_l(\lambda)\|_2$$
$$= \begin{cases} \dfrac{\gamma_l(\lambda)}{\|\gamma_l(\lambda)\|_2}, & \gamma_l(\lambda) \neq 0, \\ \{u_l : \|u_l\|_2 \leq 1\}, & \gamma_l(\lambda) = 0, \end{cases} \tag{14}$$

$$\widetilde{\rho}_l(\lambda) \in \partial\|\widetilde{\gamma}_l(\lambda)\|_2$$
$$= \begin{cases} \dfrac{\widetilde{\gamma}_l(\lambda)}{\|\widetilde{\gamma}_l(\lambda)\|_2}, & \widetilde{\gamma}_l(\lambda) \neq 0, \\ \{\widetilde{u}_l : \|\widetilde{u}_l\|_2 \leq 1\}, & \widetilde{\gamma}_l(\lambda) = 0. \end{cases} \tag{15}$$

We can transform (12), (13) into

$$\lambda \rho_l(\lambda) + \frac{\gamma_l(\lambda)}{\nu} = \frac{D\beta_l(\lambda)}{\nu}, \tag{16}$$

$$\lambda \widetilde{\rho}_l(\lambda) + \frac{\widetilde{\gamma}_l(\lambda)}{\nu} = \frac{D\beta_l(\lambda)}{\nu} + \underbrace{\left\{ -\mathrm{diag}(s)\gamma_l^* + \frac{\widetilde{A}_{\gamma_l,1}^T}{\sqrt{n_2}}\varepsilon_{2_l} \right\}}_{=:\zeta_l}, \tag{17}$$

where $\varepsilon_{2_l}$ is $l$-th group component of $\varepsilon_2$, $\widetilde{A}_{\gamma_l,1} \in \mathbb{R}^{n_2 \times g_l}$ is the first $n_2$ rows of $\widetilde{A}_{\gamma_l}$, and $\zeta_l$ is the corresponding subset of $\zeta$, where $\zeta := -\mathrm{diag}(s)\gamma^* + \frac{\widetilde{A}_{\gamma,1}^T}{\sqrt{n_2}}\varepsilon_2$. Besides, calculations on $\zeta$ shows that

$$\zeta \sim \mathcal{N}\left( -\mathrm{diag}(s)\gamma^*, \frac{1}{n_2}\mathrm{diag}(s)\{2I_m - \mathrm{diag}(s)\nu\}\sigma^2 \right). \tag{18}$$

**Lemma B.2.** *For any group $\mathcal{G}_l$, we have $\{\langle \rho_l(\lambda), \zeta_l \rangle > 0\} \subset \{W_l \le 0\}$.*

*Proof.* First, by (16) and (17), we have that for any $\lambda > 0$,

$$\lambda \rho_l(\lambda) + \frac{\gamma_l(\lambda)}{\nu} + \zeta_l = \lambda \widetilde{\rho}_l(\lambda) + \frac{\widetilde{\gamma}_l(\lambda)}{\nu}. \tag{19}$$

Since the solution path $\gamma(\lambda)$ and $\rho(\lambda)$ are right continuous, we have $\gamma_l(\lambda) = 0$ and $\|\rho_l(\lambda)\|_2 = 1$, for $\lambda = Z_l$. Then (19) reduces to

$$\lambda \rho_l(\lambda) + \zeta_l = \lambda \widetilde{\rho}_l(\lambda) + \frac{\widetilde{\gamma}_l(\lambda)}{\nu}.$$

We must have $\lambda \le \widetilde{Z}_l$, which gives us $W_l \le 0$. Proof by contradiction. When $\lambda > \widetilde{Z}_l$, we must have $\widetilde{\gamma}_l(\lambda) = 0$ and $\widetilde{\rho}_l(\lambda) = \rho_l(\lambda) + \zeta_l/\lambda$. Since $\lambda > \widetilde{Z}_l$, we have

$$1 \ge \|\widetilde{\rho}_l(\lambda)\|_2^2 = \left\| \rho_l(\lambda) + \frac{\zeta_l}{\lambda} \right\|_2^2 = \|\rho_l(\lambda)\|_2^2 + \frac{1}{\lambda^2}\|\zeta_l\|_2^2 + \frac{2}{\lambda}\langle \rho_l(\lambda), \zeta_l \rangle \ge 1 + \frac{2}{\lambda}\langle \rho_l(\lambda), \zeta_l \rangle. \tag{20}$$

When $\langle \rho_l(\lambda), \zeta_l \rangle > 0$, the right-hand side is larger than one, which gives us the contradiction. $\square$

Now, we are ready to prove Lemma B.1.

*Proof of Lemma B.1.* First, note from (18) that, for each $l \in S_0$,

$$\zeta_l \sim \mathcal{N}\left( 0, \frac{1}{n_2}\mathrm{diag}(s)\{I_{g_l} - \mathrm{diag}(s)\nu\}\sigma^2 \right).$$

Since $\rho_l(\lambda)$ is determined by $\beta(\lambda)$ and therefore is determined by the first dataset $\mathcal{D}_1$, it is independent of $\zeta_l$. Then, conditional on $\beta(\lambda)$, $\langle \rho_l(\lambda), \zeta_l \rangle$ is also a zero-mean Gaussian distribution. By Lemma B.2, we have

$$\mathbb{P}(W_l < 0) \le \mathbb{P}\{\langle \rho_l(\lambda), \zeta_l \rangle > 0\} = \frac{1}{2}.$$

This completes the proof. $\square$

## B.2. Proof of Theorem 3.3

*Proof.* Under the event $\Upsilon$, we have a reduced model for $\widetilde{y}$,

$$\widetilde{y} = A_\beta \beta_{\widehat{S}_\beta}^* + A_\gamma \gamma_{\widehat{S}_\gamma}^* + \widetilde{\varepsilon} \tag{21}$$

where $\beta_{\widehat{S}_\beta}^*$ denotes the subvector of $\beta^*$ containing only the components indexed by $\widehat{S}_\beta$, and $\gamma_{\widehat{S}_\gamma}^*$ is defined analogously as the subvector of $\gamma^*$ restricted to the index set $\widehat{S}_\gamma$. Then we can apply Theorem 3.1 to this reduced model. $\square$

**B.3. Proof of Proposition 3.5**

In this section, we will prove the model consistency problem, i.e., Proposition 3.5. We will first construct Primal-Dual Witness (PDW) (Bach, 2008; Wainwright, 2009) for Group Split Lasso, and then approach no-false-positive.

The optimal solution of the Group Split LASSO problem (2) satisfies the KKT conditions:

$$
\begin{aligned}
0 &= -(\Sigma_X + L_D)\beta(\lambda) + \frac{D^T}{\nu}\gamma(\lambda) + \left\{\Sigma_X\beta^* + \frac{X^T}{n}\varepsilon\right\}, \\
\lambda\rho(\lambda) &= \frac{D\beta(\lambda)}{\nu} - \frac{\gamma(\lambda)}{\nu},
\end{aligned}
\tag{22}
$$

where $\rho(\lambda) \in \partial\|\gamma(\lambda)\|_2$.

Now we will show that once a PDW is successfully constructed, the solution is unique. A witness set $(\widehat{\beta}^\lambda, \widehat{\gamma}^\lambda, \widehat{\rho}^\lambda) \in \mathbb{R}^p \times \mathbb{R}^m \times \mathbb{R}^m$ can be constructed as follows:

1. Set $\widehat{\gamma}_{S_0}^\lambda = 0$, and solve the following optimization problem for $(\widehat{\beta}^\lambda, \widehat{\gamma}_{S_1}^\lambda) \in \mathbb{R}^p \times \mathbb{R}^{|S_1|}$:

$$
(\widehat{\beta}^\lambda, \widehat{\gamma}_{S_1}^\lambda) := \operatorname{argmin}_{(\beta, \gamma_{S_1})} \left\{\frac{1}{2n}\|y - X\beta\|_2^2 + \frac{1}{2\nu}\|D\beta - \gamma\|_2^2 + \lambda\|\gamma_{S_1}\|_{\mathrm{group}}\right\}.
\tag{23}
$$

2. Define $\widehat{\rho}_{S_1}^\lambda = \partial\|\widehat{\gamma}_{S_1}^\lambda\|_{\mathrm{group}}$ as the subgradient of $\|\widehat{\gamma}_{S_1}^\lambda\|_{\mathrm{group}}$.

3. Solve for $\widehat{\rho}_{S_0}^\lambda \in \mathbb{R}^{S_0}$ such that the KKT conditions (22) hold, and check whether the dual feasibility condition $\left\|\widehat{\rho}_l^\lambda\right\|_2 < 1$ holds for all $l \in S_0$.

When the PDW construction succeeds, we obtain the following lemma:

**Lemma B.3.** *If the PDW construction succeeds and the optimization problem* (23) *is strictly convex, then the solution* $(\widehat{\beta}, \widehat{\gamma})$ *is the unique optimal solution of the Group Split LASSO problem.*

*Proof.* When PDW succeeds, we have $\max_{l \in S_0}\|\widehat{\rho}_l\| < 1$, so $(\widehat{\beta}, \widehat{\gamma})$ is an optimal solution, where $\widehat{\rho}_l$ is a subgradient of $\|\widehat{\gamma}_l\|_2$. Suppose $(\widetilde{\beta}, \widetilde{\gamma})$ is another optimal solution of Group Split LASSO. Define

$$
F(\beta, \gamma) = \frac{1}{2n}\|y - X\beta\|_2^2 + \frac{1}{2\nu}\|D\beta - \gamma\|_2^2.
$$

It follows that

$$
F(\widehat{\beta}, \widehat{\gamma}) + \lambda\langle\widehat{\rho}, \widehat{\gamma}\rangle = F(\widetilde{\beta}, \widetilde{\gamma}) + \lambda\|\widetilde{\gamma}\|_{\mathrm{group}},
$$

which implies

$$
F(\widehat{\beta}, \widehat{\gamma}) - \lambda\langle\widehat{\rho}, \widetilde{\gamma} - \widehat{\gamma}\rangle - F(\widetilde{\beta}, \widetilde{\gamma}) = \lambda(\|\widetilde{\gamma}\|_{\mathrm{group}} - \langle\widehat{\rho}, \widetilde{\gamma}\rangle).
$$

From (22), we have $\frac{\partial F(\widehat{\beta}, \widehat{\gamma})}{\partial\widehat{\beta}} = 0$ and $\frac{\partial F(\widehat{\beta}, \widehat{\gamma})}{\partial\widehat{\gamma}} = -\lambda\widehat{\rho}$. Therefore,

$$
F(\widehat{\beta}, \widehat{\gamma}) + \left\langle\frac{\partial F(\widehat{\beta}, \widehat{\gamma})}{\partial\widehat{\gamma}}, \widetilde{\gamma} - \widehat{\gamma}\right\rangle + \left\langle\frac{\partial F(\widehat{\beta}, \widehat{\gamma})}{\partial\widehat{\beta}}, \widetilde{\beta} - \widehat{\beta}\right\rangle - F(\widetilde{\beta}, \widetilde{\gamma}) = \lambda(\|\widetilde{\gamma}\|_{\mathrm{group}} - \langle\widehat{\rho}, \widetilde{\gamma}\rangle).
$$

Since $F$ is convex, the left-hand side of the above equation is non-positive. That means,

$$
\sum_l \|\widetilde{\gamma}_l\|_2 \le \sum_l \langle\widetilde{\rho}_l, \widetilde{\gamma}_l\rangle.
$$

Since $\|\widehat{\rho}_l\| < 1$, to make the above inequality holds, we must have $\widetilde{\gamma}_l = 0$ for any $l \in S_0$. This implies that $(\widetilde{\beta}, \widetilde{\gamma})$ is also an optimal solution of problem (23), which contradicts the fact that $F$ is strictly convex. Therefore, $(\widehat{\beta}, \widehat{\gamma})$ is the unique solution of Group Split LASSO. $\qquad\square$

Next, we will show how to derive the $\nu$-incoherence condition and apply it to the proof of Theorem 3.5.

From equation (22), the solution $(\widehat{\beta}, \widehat{\gamma})$ of (2) satisfies

$$\lambda \nu \widehat{\rho} = -H_\nu(\widehat{\gamma} - \gamma^*) + \omega,$$

where $\omega = D[\Sigma_X + L_D]^{-1} \frac{X^T}{n} \varepsilon$. Using the definitions of $H_\nu^{00}$, $H_\nu^{11}$, $H_\nu^{10}$, and $H_\nu^{01}$, we obtain

$$\lambda \nu \begin{bmatrix} \widehat{\rho}_{S_1} \\ \widehat{\rho}_{S_0} \end{bmatrix} = - \begin{bmatrix} H_\nu^{11} & H_\nu^{10} \\ H_\nu^{01} & H_\nu^{00} \end{bmatrix} \begin{bmatrix} \widehat{\gamma}_{S_1} - \gamma_{S_1}^* \\ 0_{S_0} \end{bmatrix} + \begin{bmatrix} \omega_{S_1} \\ \omega_{S_0} \end{bmatrix},$$

which yields

$$\lambda \nu \widehat{\rho}_{S_1} = -H_\nu^{11}(\widehat{\gamma}_{S_1} - \gamma_{S_1}^*) + \omega_{S_1},$$
$$\lambda \nu \widehat{\rho}_{S_0} = -H_\nu^{01}(\widehat{\gamma}_{S_1} - \gamma_{S_1}^*) + \omega_{S_0}.$$

Since $H_\nu^{11}$ is invertible, we can solve for $\widehat{\gamma}_{S_1} - \gamma_{S_1}^*$ and obtain

$$\widehat{\rho}_{S_0} = H_\nu^{01}[H_\nu^{11}]^{-1}\widehat{\rho}_{S_1} + \frac{1}{\lambda \nu}\{\omega_{S_0} - H_\nu^{01}[H_\nu^{11}]^{-1}\omega_{S_1}\}.$$

Therefore, for any $l \in S_0$,

$$\|\widehat{\rho}_l\| \leq \|H_\nu^{l1}[H_\nu^{11}]^{-1}\widehat{\rho}_{S_1}\| + \frac{1}{\lambda \nu}\|\omega_l - H_\nu^{l1}[H_\nu^{11}]^{-1}\omega_{S_1}\|, \tag{24}$$

where $H_\nu^{l1}$ denotes the submatrix corresponding to group $G_l$.

By the $\nu$-incoherence condition, we have

$$\begin{aligned} \|H_\nu^{l1}[H_\nu^{11}]^{-1}\widehat{\rho}_{S_1}\| &\leq \|H_\nu^{l1}[H_\nu^{11}]^{-1}\|_2\|\widehat{\rho}_{S_1}\| \\ &\leq \frac{1 - \chi_\nu}{\sqrt{|S_1|}}\sqrt{|S_1|} \\ &= 1 - \chi_\nu, \quad \text{for any } l \in S_0. \end{aligned} \tag{25}$$

Meanwhile, since $\omega = D[\Sigma_X + L_D]^{-1}\frac{X^T}{n}\varepsilon$, it follows that

$$\begin{aligned} \|\omega_l - H_\nu^{l1}[H_\nu^{11}]^{-1}\omega_{S_1}\| &\leq (1 + \|H_\nu^{l1}[H_\nu^{11}]^{-1}\|)\|\omega\| \\ &\leq \left(\frac{1 - \chi_\nu}{\sqrt{|S_1|}} + 1\right)\|\omega\| \\ &\leq \frac{(\sqrt{|S_1|} + 1 - \chi_\nu)\kappa}{\sqrt{n|S_1|}}\|\varepsilon\|. \end{aligned} \tag{26}$$

We now invoke a concentration inequality for norms to establish a tail bound:

$$\mathbb{P}\left\{\left|\|\varepsilon\| - \sqrt{m}\right| \geq t\right\} \leq 2\exp\left(\frac{-c_1 t^2}{M^4}\right), \quad \text{for all } t \geq 0,$$

where $M := \max_i \|\varepsilon_i\|_{\psi_2}$ and $c_1$ are non-negative constants. Choosing $t = M^2\sqrt{\frac{m}{c_1}}$, we obtain

$$\mathbb{P}\left\{\|\varepsilon\| \geq \left(\frac{M^2}{\sqrt{c_1}} + 1\right)\sqrt{m}\right\} \leq e^{-m}.$$

Thus, let $C = (\frac{M^2}{\sqrt{c_1}} + 1)$, and take

$$\lambda = \lambda_n \geq \frac{4(\sqrt{|S_1|} + 1 - \chi_\nu)C\sigma}{\sqrt{|S_1|}\nu\chi_\nu}\sqrt{\frac{m}{n}}, \tag{27}$$

then

$$\frac{(\sqrt{|S_1|} + 1 - \chi_\nu)\kappa}{\lambda\nu\sqrt{n|S_1|}}\|\varepsilon\| \le \frac{\chi_\nu}{4} \tag{28}$$

holds with probability at least $1 - e^{-m}$.

Combining the results of inequalities (24)–(28), we conclude that when $\{\lambda_n\}$ satisfies

$$\frac{4(\sqrt{|S_1|} + 1 - \chi_\nu)C\sigma}{\sqrt{|S_1|}\nu\chi_\nu}\sqrt{\frac{m}{n}} \le \lambda_n,$$

then

$$\mathbb{P}\left\{\text{for all } l \in S_0, \|\widehat{\rho}_l\| \le 1 - \frac{3\chi_\nu}{4}\right\} > 1 - e^{-m}.$$

This shows that $\widehat{\gamma}$ will not produce false positives with probability greater than $1 - e^{-m}$.

To show that $\widehat{\gamma}$ recovers $S_1$, it is sufficient to show the elementwise sign-consistency, *i.e.*, $\text{sign}(\widehat{\gamma}_i) = \text{sign}(\gamma_i^*)$ for each $i \in [m]$. Take $\lambda = \lambda_n$ in (27) and consider the infinity norm on both sides. There holds

$$\left\|\widehat{\gamma}_{S_1} - \gamma_{S_1}^*\right\|_\infty \le \lambda_n\nu\left\|[H_\nu^{11}]^{-1}\right\|_\infty + \left\|[H_\nu^{11}]^{-1}\omega_{S_1}\right\|_\infty. \tag{29}$$

Note that the first term on the right-hand side is deterministic; therefore, we only focus on the second term. Recall that

$$\omega = D\left[\Sigma_X + L_D\right]^{-1}\frac{X^\top}{n}\varepsilon.$$

Therefore, for each $i$,

$$\frac{\omega_i}{\nu} = \frac{1}{\nu}D_{i,\cdot}\left[\Sigma_X + L_D\right]^{-1}\frac{X^\top}{n}\varepsilon$$

is a Gaussian random variable with variance

$$\begin{aligned}
\text{Var}\left(\frac{\omega_i}{\nu}\right) &= \frac{1}{\nu^2}D_{i,\cdot}\left[\Sigma_X + L_D\right]^{-1}\frac{X^\top X}{n^2}\left[\Sigma_X + L_D\right]^{-1}D_{i,\cdot}^\top\sigma^2 \\
&= \frac{\sigma^2}{\nu n}D_{i,\cdot}\left[\Sigma_X + L_D\right]^{-1}\Sigma_X\left[\Sigma_X + L_D\right]^{-1}D_{i,\cdot}^\top \\
&\le \frac{\sigma^2}{\nu n}D_{i,\cdot}\left[\Sigma_X + L_D\right]^{-1}D_{i,\cdot}^\top \le \frac{\sigma^2}{n}.
\end{aligned}$$

By the standard Gaussian tail inequality, there holds

$$\mathbb{P}\left(\left\|[H_\nu^{11}]^{-1}\omega_{S_1}\right\|_\infty > \nu t\right) \le 2|S_1|\exp\left(-\frac{n}{2\sigma^2}t^2 C_{\min}^2\right).$$

Take $t = \frac{\sigma}{C_{\min}}\sqrt{\frac{2m}{n}}$,, we have

$$\mathbb{P}\left(\left\|[H_\nu^{11}]^{-1}\omega_{S_1}\right\|_\infty > \frac{\nu\sigma}{C_{\min}}\sqrt{\frac{2m}{n}}\right) \le \exp(-m + \log(2|S_1|)) \le e^{-m},$$

Therefore, with probability at least $1 - e^{-m}$,

$$\left\|\widehat{\gamma}_{S_1} - \gamma_{S_1}^*\right\|_\infty \le \frac{\nu\sigma}{C_{\min}}\sqrt{\frac{2m}{n}} + \lambda_n\nu\left\|(H_\nu^{11})^{-1}\right\|_\infty. \tag{30}$$

If $\min_{i \in S_1}\gamma_i^* > \frac{\nu\sigma}{C_{\min}}\sqrt{\frac{2m}{n}} + \lambda_n\nu\left\|(H_\nu^{11})^{-1}\right\|_\infty$, then $\widehat{\gamma}$ exactly recovers the support of $\gamma^*$ with probability at least $1 - e^{-m}$.

## C. Split Group Knockoff in Multivariate Response

In this section, we consider the matrix-valued linear model

$$Y = XB^* + E, \qquad \Gamma^* = DB^*, \tag{31}$$

where $Y \in \mathbb{R}^{n \times d}$ is the response matrix, $X \in \mathbb{R}^{n \times p}$ is the design matrix, $B^* \in \mathbb{R}^{p \times d}$ is the unknown coefficient matrix, and $E \in \mathbb{R}^{n \times d}$ is a noise matrix whose rows are independent and satisfy $\text{vec}(E) \sim \mathcal{N}(0, \sigma^2 I_{nd})$. We introduce $\Gamma^* \in \mathbb{R}^{m \times d}$ as the transformed parameter matrix, defined by a pre-specified transformation matrix $D \in \mathbb{R}^{m \times p}$. We will provide the method based on matrix version.

We assume that $\Gamma^*$ exhibits a sparse group structure across its rows. Specifically, the row indices of $\Gamma^*$ are partitioned into $k$ disjoint groups,

$$\mathbb{G}_1 \cup \mathbb{G}_2 \cup \cdots \cup \mathbb{G}_k = \{1, 2, \ldots, m\},$$

where $\mathbb{G}_l$ denotes the index set of the $l$-th group and $g_l := |\mathbb{G}_l|$ is its group size. Let

$$\Gamma_l^* := \Gamma^*[\mathbb{G}_l, :] \in \mathbb{R}^{g_l \times d}$$

denote the submatrix corresponding to the $l$-th group. We assume that only a small fraction of these groups contains nonzero signals. Accordingly, define the true null and non-null group sets as

$$S_0 := \{l : \Gamma_l^* = 0_{g_l \times d}\}, \qquad S_1 := \{l : \Gamma_l^* \neq 0\}.$$

Let $\widehat{S} \subset \{1, 2, \ldots, k\}$ denote an estimated support set. Our goal is to recover $\widehat{S}$ while controlling the group-wise false discovery rate (FDR) at a target level $q$, defined as

$$\text{FDR}_{\text{group}} = \mathbb{E}\left[ \frac{\left| \widehat{S} \cap S_0 \right|}{\left| \widehat{S} \right| \vee 1} \right].$$

*Application to outlier detection among large language models.* In the task of identifying outliers among $R$ large language models, we assume that each model $r \in [R]$ admits an associated coefficient matrix $B^{r,*} \in \mathbb{R}^{\tilde{p} \times d}$ with $\tilde{p} = p/R$, which characterizes its predictive behavior. Without loss of generality, we take the first model as the reference inlier. For $r \geq 2$, define

$$\Gamma^{r-1,*} := B^{r,*} - B^{1,*},$$

which measures the deviation of the $r$-th model from the reference model. The transformation matrix $D \in \mathbb{R}^{p(R-1) \times p}$ is constructed such that $\Gamma^* = DB^*$ stacks these pairwise deviations.

### C.1. Model Framework for Split Group Knockoff

To facilitate stable estimation and group-level inference for the group-sparse transformed parameter $\Gamma^*$, we relax the hard constraint $\Gamma = DB$ by introducing a quadratic penalty. This leads to the following matrix-valued group-lasso optimization problem:

$$\min_{B, \Gamma} \frac{1}{2n} \|Y - XB\|_F^2 + \frac{1}{2\nu} \|DB - \Gamma\|_F^2 + \lambda \sum_{l=1}^{k} \|\Gamma_l\|_F, \quad \lambda > 0, \tag{32}$$

where $\|\cdot\|_F$ denotes the Frobenius norm and $\nu > 0$ controls the proximity between $DB$ and $\Gamma$.

To construct valid knockoffs, we randomly split the dataset $\mathcal{D} = (X, Y)$ into two independent subsets $\mathcal{D}_1 = (X_1, Y_1)$ and $\mathcal{D}_2 = (X_2, Y_2)$ with sample sizes $n_1$ and $n_2$, respectively, where $n_1 + n_2 = n$ and $n_2 \geq m + p$.

Using $\mathcal{D}_1$, we estimate the regression coefficient matrix $B(\lambda)$, either along a regularization path or via cross-validation.

Next, we construct knockoff features using $\mathcal{D}_2$. Define the augmented matrices

$$\widetilde{Y} = \begin{pmatrix} Y_2/\sqrt{n_2} \\ 0_{m \times d} \end{pmatrix}, \qquad A_B = \begin{pmatrix} X_2/\sqrt{n_2} \\ D/\sqrt{\nu} \end{pmatrix},$$

$$A_\Gamma = \begin{pmatrix} 0_{n_2 \times m} \\ -I_m/\sqrt{\nu} \end{pmatrix}, \qquad \widetilde{E} = \begin{pmatrix} E_2/\sqrt{n_2} \\ 0_{m \times d} \end{pmatrix}. \tag{33}$$

With these definitions, the augmented linear model can be written as

$$\widetilde{Y} = A_B B^* + A_\Gamma \Gamma^* + \widetilde{E}.$$

We then construct a split knockoff copy $\widetilde{A}_\Gamma$ satisfying

$$
\begin{aligned}
\widetilde{A}_\Gamma^\top \widetilde{A}_\Gamma &= A_\Gamma^\top A_\Gamma, \\
A_B^\top \widetilde{A}_\Gamma &= A_B^\top A_\Gamma, \\
A_\Gamma^\top \widetilde{A}_\Gamma &= A_\Gamma^\top A_\Gamma - \operatorname{diag}(s),
\end{aligned}
\tag{34}
$$

where $s \in \mathbb{R}_+^m$.

Given the estimator $B(\lambda)$ obtained from $\mathcal{D}_1$, we solve the following group-lasso problem:

$$
\begin{aligned}
\Gamma_l(\lambda) := \arg\min_{\Gamma_l} \frac{1}{2} &\left\| \widetilde{Y} - A_B B(\lambda) - \sum_{l=1}^k A_{\Gamma_l} \Gamma_l \right\|_F^2 \\
&+ \lambda \sum_{l=1}^k \|\Gamma_l\|_F,
\end{aligned}
\tag{35}
$$

where $A_{\Gamma_l} \in \mathbb{R}^{(n_2+m)\times g_l}$ denotes the columns of $A_\Gamma$ corresponding to group $\mathbb{G}_l$. We define the feature significance of group $\mathbb{G}_l$ as

$$Z_l := \sup\{\lambda : \Gamma_l(\lambda) \neq 0\}.$$

Replacing $A_\Gamma$ with its split knockoff copy $\widetilde{A}_\Gamma$, we similarly define

$$
\begin{aligned}
\widetilde{\Gamma}_l(\lambda) := \arg\min_{\widetilde{\Gamma}_l} \frac{1}{2} &\left\| \widetilde{Y} - A_B B(\lambda) - \sum_{l=1}^k \widetilde{A}_{\Gamma_l} \widetilde{\Gamma}_l \right\|_F^2 \\
&+ \lambda \sum_{l=1}^k \|\widetilde{\Gamma}_l\|_F.
\end{aligned}
\tag{36}
$$

The knockoff significance of group $\mathbb{G}_l$ is defined as

$$\widetilde{Z}_l := \sup\{\lambda : \widetilde{\Gamma}_l(\lambda) \neq 0\}.$$

We then construct the split knockoff statistic

$$W_l := Z_l \cdot \operatorname{sign}(Z_l - \widetilde{Z}_l), \qquad l = 1, \ldots, k.$$

Given a target FDR level $q \in (0, 1)$, the selected groups are defined as

$$\widehat{S} := \{l : W_l \geq T_q\},$$

where $T_q$ is the group split knockoff (or split knockoff+) threshold defined same in (6).

We can also give the FDR control theorem for multivariate response problem.

**Theorem C.1** (FDR Control of Multivariate Response Problem). *For all $0 < q \leq 1$ and all $\nu > 0$, the following holds:*

*(i) (**Modified group-FDR of Split Group Knockoff**).*

$$\mathbb{E}\left[ \frac{\left| \left\{ l : l \in \widehat{S} \cap S_0 \right\} \right|}{\left| \widehat{S} \right| + q^{-1}} \right] \leq q.$$

| Subject | Entropy | Model Accuracies | | | | | |
|---------|---------|------------------|---------------------|-------|-----------|---------------|---------------|
| | | DeepSeek-v3.2 | DeepSeek-v3.2-Speciale | GPT-5 | Qwen-Huge | Qwen-Instruct | Qwen-Thinking |
| **Law** | 1.0314 | 58.9% | 57.0% | 65.6% | 39.9% | 37.0% | 44.6% |
| **Chemistry** | 1.0361 | 76.8% | 53.9% | 72.0% | 29.2% | 60.1% | 64.0% |
| **Business** | 0.9988 | 73.0% | 61.3% | 69.6% | 34.3% | 51.5% | 59.3% |
| **Engineering** | 1.3493 | 60.9% | 36.7% | 63.5% | 35.3% | 39.1% | 46.5% |
| **Philosophy** | 0.9740 | 62.0% | 64.2% | 74.3% | 34.1% | 38.0% | 38.5% |
| **Physics** | 1.0324 | 71.9% | 50.0% | 72.7% | 35.2% | 63.1% | 65.8% |
| **Health** | 0.9709 | 55.7% | 51.4% | 51.4% | 40.3% | 45.8% | 50.6% |
| **Comp. Sci.** | 1.0247 | 66.7% | 58.3% | 55.2% | 41.7% | 45.8% | 57.3% |
| **Math** | 0.9257 | 77.1% | 60.9% | 75.2% | 25.2% | 69.4% | 69.4% |
| **Biology** | 0.8916 | 71.3% | 56.5% | 72.2% | 40.7% | 49.1% | 58.3% |
| **History** | 0.9283 | 51.9% | 48.9% | 59.3% | 42.2% | 31.9% | 43.0% |
| **Economics** | 0.8887 | 63.5% | 57.1% | 67.6% | 40.6% | 46.5% | 54.1% |
| **Psychology** | 0.9445 | 57.4% | 57.9% | 64.1% | 44.6% | 41.0% | 48.2% |

*Table 4.* Average Entropy and Individual Model Accuracies by Task Subject.

*(ii)* **(group-FDR of Split Group Knockoff+).**

$$\mathbb{E}\left[\frac{\left|\left\{l : l \in \widehat{S} \cap S_0\right\}\right|}{\left|\widehat{S}\right| \vee 1}\right] \leq q.$$

The proof follows exactly the same arguments as that of Theorem 3.1. The only modifications are notational: vector-valued quantities are replaced by their matrix-valued counterparts, and all $\ell_2$ norms are replaced by Frobenius norms. No other changes are required.

## D. Details on Data Filtering and Thresholding

Since questions where all models reach a consensus provide negligible signal for distinguishing model-specific behavioral anomalies, we implement a data filtering step to maximize the discriminative power of our analysis. We quantify the **response divergence** for each question using Shannon entropy calculated over the distribution of model responses. A pre-defined threshold $\tau = 0.5$ is applied to the normalized entropy score to retain only those samples where models exhibit significant disagreement.

This filtering process yields a dataset of contentious questions used for the main experiments reported in the paper. The specific sample sizes for each task are: Law (616), Engineering (493), Physics (366), Chemistry (336), Math (258), Health (253), Business (204), Psychology (195), Philosophy (179), Economics (170), History (135), Biology (108), and Computer Science (96).

## E. Details Results for Section 5

In this section, we will list the detailed results for LLM behavior auditing experiments. Table 4 shows all model accuracies within each task subject.

