# OpenReview forum: "Split Group Knockoffs: Controlling False Discovery Rate in Transformational Group Sparsity"
_ICML.cc/2026/Conference — ICML 2026 regular_

### Official Review · Reviewer_g42H · 2026-03-12

**Soundness:** 2
**Presentation:** 3
**Significance:** 2
**Originality:** 3
**Overall Recommendation:** 4
**Confidence:** 3

**Summary:**

This work extends the knockoffs variable selection framework to handle transformational group sparsity. Rather than trying to identify truly important input variables, this work aims to identify groups that are important after transformation, supporting questions like "is model A's behavior different from model B?" It is proven that this method provides valid false discovery rate control. The method is demonstrated with two case studies, one studying LLM behavior and another studying which brain regions/connections matter in the development of Alzheimer's disease.

**Compliance With Llm Reviewing Policy:**

Affirmed.

**Final Justification:**

After some constructive back and forth with the authors, I am increasing my score to a 4, although my opinion is closer to a 3.5 (true borderline). I think this work would be much stronger with some more time to revise/include alternative case studies, but it could reasonably cross the threshold for acceptance with smaller changes.

I continue to believe that the impact/scope of this work is somewhat limited. This is for two primary reasons:
1) The theoretical guarantees of this work only hold when the modeling choice of using a linear model is correct. In future iterations of this work, this could be addressed by considering non/semi-parametric or broadly more flexible models. However, this constraint is not new to this work, and is fairly common across knockoffs work.
2) This work does not, in my opinion, contain good real-world applications of the framework. In the rebuttal, the authors suggested a more compelling application in genomics. This will probably not be added by the camera-ready deadline, but I believe some discussion along these lines will help sell the work.

Overall, I broadly maintain my primary concern, but feel that the authors have suggested some avenues to help improve the impact of the work and therefore blunted the concern a little.

**Key Questions For Authors:**

- What are some real world settings where the linearity assumption behind this work is likely to be met, and the key question of interest cannot be solved without split group knockoffs? In my opinion, this is not answered well in the current work. If answered well this would better demonstrate the impact of this approach.

**Limitations:**

Yes

**Strengths And Weaknesses:**

Strengths
- Presentation
  - I found the presentation in this work clear and easy to follow. Related literature is cited appropriately.
- Originality
  - Transformational group sparsity is, to my knowledge, a new addition to the literature on knockoffs. While both of the key components of this work are well studied, their combination is new and worth pursuing.

Weaknesses
- My primary critique of this work is that it is not particularly well motivated, and I do not find the current case studies appropriate for it. I discuss this in more detail below, but this is the main factor decreasing my score.
- Soundness
  - In applying this method to LLMs, it is assumed that "each model admits an associated coefficient vector, which encodes its predictive behavior.” This is assuming that each of the LLMs studied are linear models that map between two 64 dimensional text embeddings, which is generally untrue.
  - For theorem 3.3, assuming that variable filtering can be performed until p < n with zero false negatives is quite strong.

- Significance
  - The significance of this work is limited by the fact that it assumes labels come from a linear model. This limits the applicability of the proposed framework and theory.
  - This iteration of the work does not articulate the need for this extension particularly well, and this issue is highlighted by the case studies. As discussed above, the LLM setting is not really appropriate. The case study in Section 6 may be better motivated, but the brain region section experiment sets $D=I$; in my understanding, this means the exact same problem could have been solved by the original knockoffs method. Section 6.2 seems to be the only evaluation that is both valid and a new capacity introduced by this work.

- Presentation (Minor)
  - While the writing was generally clear, I do want to raise a couple easily fixed suggestions for improvement:
    - Currently, the beginning of section 3 contains specific references/definitions relating to the experiment in section 5, such as describing the matrix D for that setting. I recommend removing this discussion until later, since it is a little confusing as is.
    -  Section 4 states: “Let 0 = s0 < s1 < · · · < sK = p denote a partition of {1, . . . , p} into K contiguous blocks, where s1 − s0 = 10 and sk − sk−1 = 5" — is s the size of each partition, or an actual set? Or indices? Clarifying this would be helpful
    - The matrix B, which is referenced in section 5, does not seem to be defined until much later

---

> ### Author Rebuttal · Authors · 2026-03-30
>
> We thank the reviewer for the detailed and constructive feedback. We will address the concerns below and correct for the presentation issues in the updated version.
>
> **Question.** On the linear model assumption and LLM application.
>
> **Response.** We thank the reviewer for this comment. The linear model in our paper is used as a working model, rather than as a claim that the true data-generating mechanism is exactly linear. Since the true model is unknown, exact linearity cannot, in general, be validated or invalidated from the data, and such working-model formulations are standard in the selective inference and knockoff literature. Our main contribution is not a new linear model, but a new inferential framework for grouped transformational sparsity, namely FDR-controlled inference on $\gamma=D\beta$ when $\gamma$ has a group structure. This problem setting, introduced in Section 3, is not addressed by existing methods, regardless of the underlying predictive model. The linear setup provides a clean foundation for developing the method and proving the groupwise FDR guarantees.
>
> In the LLM application, we do not assume that the raw input–output mapping of an LLM is itself linear. Instead, we use a linear model on top of embeddings as a first-order approximation to capture dominant patterns in behavior. This is a standard representation-level paradigm in recent LLM interpretability [1-4]. We will explicitly clarify this modeling perspective in the revision to avoid potential misunderstanding.
>
> **References**
>
> [1] Park et al. (2023). The linear representation hypothesis and the geometry of large language models.
>
> [2] Marks, S., & Tegmark, M. (2023). The geometry of truth: Emergent linear structure in large language model representations of true/false datasets.
>
> [3] Arditi et al. (2024). Refusal in language models is mediated by a single direction. Advances in Neural Information Processing Systems.
>
> [4] Tomihari, A., & Sato, I. (2024). Understanding linear probing then fine-tuning language models from ntk perspective.
>
> **Question.** About the applicational scope and contribution of SGK.
>
> **Response.** The linear model is widely applicable to various fields involving structured data, including genomic analysis, econometrics, and neuroimaging. Additionally, it can serve as a working model or linear transformation for embeddings of nonstructured data, thanks to its high interpretability. Such settings often involve natural group structures (e.g., gene pathways, spatial regions, or feature blocks).
>
> In particular, when the goal is to identify representation differences under a known group structure (i.e., inference on grouped transformed effects $\gamma = D\beta$), existing methods are not applicable, making our framework necessary.
>
>
> **Question.**  About the screening assumption in Theorem 3.3.
>
> **Response.** The screening assumption is required to ensure that $n>m+p$. Methods such as DC-SIS procedures [1] provide such guarantees with high probability under suitable conditions.
>
> **References**
>
> [1] Li et al. (2012). Feature Screening via Distance Correlation Learning. Journal of the American Statistical Association.
>
> **Question.**  Presentation issues.
>
> **Response.**  We thank the reviewer for these helpful suggestions and will revise accordingly.

---

> > ### Author Rebuttal · Reviewer_g42H · 2026-04-01
> >
> > I thank the authors for this helpful response.
> >
> > The point about identifying representation differences under known group structures is well taken, and I appreciate this elaboration. In future work, I think a case study on a domain like genomics, where the known group structure is of interest, would really help show the value of this work.
> >
> > **On the linear model assumption and LLM application.**
> >
> > We may need to agree to disagree on this point, but I have responded to the author's position below in case they would like to respond again. Currently, I remain fairly unconvinced.
> >
> > > The linear model in our paper is used as a working model, rather than as a claim that the true data-generating mechanism is exactly linear.
> >
> > Since the target for inference is defined in terms of coefficients from a linear model with 0-centered Gaussian noise, I do see this as a claim that the true data-generating mechanism is linear. Either this assumption is being made, or the parameter of interest is not necessarily connected to the true data-generating mechanism, which makes it a bit of an odd target to me. This said, I acknowledge that this assumption is quite prevalent across work on knockoffs, and is not new to this work.
> >
> > > In the LLM application... we use a linear model on top of embeddings as a first-order approximation to capture dominant patterns in behavior. This is a standard representation-level paradigm in recent LLM interpretability [1-4].
> >
> > My understanding of this line of work is that it assumes/studies linearity from the (high dimensional) _representation space_ of the LLM, not the input text embedding. As such, [1-4] do not assume the entire LLM is linear directly from the Matryoshka embedding, but rather that some sub-component of the LLM is. Thus, I recognize that treating the input-output mapping as linear is a modeling choice, but I maintain that it comes with a strong implicit assumption of linearity if we are going to use it to draw conclusions about the actual LLM.

---

> > > ### Author Response · Authors · 2026-04-02
> > >
> > > **Response.**
> > >
> > > We thank the reviewer for the thoughtful follow-up and for the positive comments on the grouped-structure setting. We also appreciate the suggestion about genomics; we agree that applications with scientifically meaningful group structures would be a valuable direction for future work.
> > >
> > > **On the linear model assumption.**
> > > Our intention is not to claim that the true data-generating mechanism is literally linear with Gaussian noise. Rather, the linear-Gaussian model is the setting under which we develop the methodology and establish the theoretical guarantee. This is the standard role of an explicit statistical model: it provides a tractable and interpretable framework in which the target parameter, procedure, and theory can be clearly defined. The fact that the theory is proved under this model does not mean the method becomes meaningless whenever the real system is not exactly generated from that model; it only means that this is the regime under which the guarantee is formally established.
> > >
> > > In practice, one never knows the true data-generating mechanism, and any model-based analysis necessarily involves approximation and modeling choices that can be questioned in principle. No statistical model fully captures reality. Our position is that the value of the model in real analysis lies in whether it reveals interpretable phenomena in data. In our case, the linear formulation provides such a framework, and through it we are able to identify structured and interpretable signals that would not be directly accessible otherwise. We therefore do not view this target as an odd one; rather, it is a standard model-based inferential target, explicitly defined under a chosen statistical framework and useful insofar as it captures meaningful regularities in the observed data. The reviewer’s suggestion of a future genomics case study further highlights the broad applicability and practical relevance of our framework, especially in domains where scientifically meaningful group structures arise naturally.
> > >
> > > **On the LLM application.**
> > > We would like to clarify that the references cited in our previous response were not meant to claim that our setup is identical to hidden-state-based interpretability work. Rather, they were intended to illustrate a narrower point: linear models are a very common and well-accepted modeling choice in LLM-related studies.
> > >
> > > In our setting, we model the relationship between input and output representations through a linear map on a shared semantic embedding space. This is the framework we use for behavior auditing. Our goal is not to assert that the full internal mechanism of an LLM is literally linear, but to adopt a tractable approximation for characterizing cross-model behavioral differences in a common representation space. This choice is empirically meaningful in our setting: under this linear representation model, we identify coherent and interpretable subject-level deviations that align with patterns reported in prior studies, supporting the practical usefulness of this approximation.
> > >
> > > More broadly, as in the previous point, the true relationship between LLM inputs and outputs is inherently black-box, and there is no principled way to certify that one particular statistical model is the exact ground truth. This is precisely why model-based analysis relies on tractable approximations. Our claim is therefore not that the linear model is the uniquely correct description of the LLM, but that it provides a useful and analyzable framework in which meaningful behavioral patterns can be discovered.

---

### Official Review · Reviewer_hSWt · 2026-03-12

**Soundness:** 3
**Presentation:** 2
**Significance:** 2
**Originality:** 3
**Overall Recommendation:** 4
**Confidence:** 4

**Summary:**

This work introduces the Split Group Knockoff (SGK) framework, which provides a principled approach to variable selection under group sparsity with false discovery rate control within the analysis framework.
SGK builds on split knockoffs to group transformed variables, offering more flexibility when dealing with advanced data structures.
It offers statistical guarantees similar to those of the traditional Knockoff framework.
Additionally, two empirical studies are provided: one on LLM auditing and one on multimodal brain connectivity.
These studies demonstrate that SGK identifies meaningful group-level signals while controlling for false discoveries.

**Compliance With Llm Reviewing Policy:**

Affirmed.

**Final Justification:**

I found the contribution interesting, but the rebuttal did not bring any additional insight. There remain quite a few gap to make the proposed approach a viable solution.  I don't know who would use it as-is.
So I keep my "weak accept".

**Key Questions For Authors:**

* Theorem 3.3 relies on the hypothesis that the estimated support set contains the true support set, which is highly unlikely. Assumption 3.4 (nu-incoherence) is impossible to check and is likely violated in most empirical problems.
* It seems to me that the authors implicitly assume that the groups are consistent with the support, either fully within or outside of it. This is not generally true and warrants at least a word of caution.
* The authors seem to use the emergence time of groups. However, it has been shown for KOs that the Lasso coefficient difference was a better choice. Therefore, the current SGK procedure is likely suboptimal.
* The authors also seem to consider the empirical covariance sufficient for knockoff creation (Equation 4). In practice, however, it is very fragile, and there is no clear guarantee of exchangeability. This must be acknowledged.
* In their simulations, the authors use favorable settings (n >> p, mild correlation). It would be interesting to investigate how the estimator behaves under harder, more realistic settings.
* The SNR in the simulations should be described properly.
* The choice of Nu is unclear; it appears that the authors cherry-pick "good values," which leads to overfitting. Sample splitting is statistically inefficient and costly in the case of cross-validation.
* Fig. 1 is not particularly clear. As someone unfamiliar with LLM auditing, I have a hard time understanding the details. Additionally, the mini-caption is unhelpful.
* Fig. 2 indicates nominal q in the caption.
* Fig. 2 has extremely small text, making it almost impossible to understand.
* As a neuroimaging expert, I find the neuroimaging experiment uninteresting.: How do we interpret the numbers in Table 2? Do they correspond to different significance levels? Pooling data from a 1.5T and a 3T machine is not a significant achievement, especially since the 3T data is almost certainly more informative.
* Why are the sample splitting schemes different for region selection and connectome selection?

**Limitations:**

* Overall, many problems arise when one wants to use such procedures in real world settings (high correlation in the design, low sample/dimension ratio, randomness of the knowckoff outcome, hyperparameter setting). All these are mostly if not completely missed by the current submission.
* besides using a standard q values, such as  q=0.05 (the standard for FDR control in many fields), then the procedure is often powerless and unstable. Moreover, SGK+ requires a  large support (SGK is particularly sloppy in such a case).  This is not mentioned.

**Strengths And Weaknesses:**

Strengths

* I think the idea has strong potential because it is versatile.
* The guarantees offered are meaningful in the SGK+ case (see Theorem 3.1(ii)).
* Experiments are carried out in two different real-data settings with relatively good face validity.

Weaknesses

* The authors propose SGK inference (Theorem 3.1(i)), which is not a good choice because it is difficult to interpret and misleading to users. The paper should only use the procedure in Theorem 3.1(ii) and mention the former as a complement.
* Setting nu is non-trivial.
* The theoretical results rely on unrealistic and unverifiable assumptions.
* There is no clear baseline method for comparison.

---

> ### Author Rebuttal · Authors · 2026-03-30
>
> We thank the reviewer for the detailed and constructive feedback. We address the concerns below.
>
> **Question.** About the screening assumption and Assumption 3.4 in Theorem 3.3.
>
> Response.These assumptions are standard in high-dimensional variable selection. The screening assumption was similarly used in [1] and can be achieved by standard methods such as DC-SIS [2]. Assumption 3.4 ($\nu$-incoherence) is an identifiability condition, analogous to incoherence/irrepresentable conditions in Lasso-type analyses [3], and is mainly needed for power. Importantly, neither assumption is required for FDR control when $n>p$.
>
> [1] Cao et al. (2024). Controlling the false discovery rate in transformational sparsity: Split knockoffs.
>
> [2] Li et al. (2012). Feature Screening via Distance Correlation Learning.
>
> [3] Zhao and Yu (2006). On Model Selection Consistency of Lasso.
>
> **Question.** About the implicit assumption of group sparsity.
>
> Response. We would like to clarify that this is exactly the defining feature of group sparsity: the group, rather than the individual coordinate, is the unit of inference. Thus, a group is selected if its coefficient subvector is nonzero, and excluded otherwise. This all-in/out behavior is therefore not an extra assumption, but the intended modeling target. If one wishes to allow partial selection within a group, then a different structural model, such as sparse-group sparsity, would be more appropriate.
>
> **Question.** Why not use the Lasso coefficient difference?
>
> Response. Our setting differs from classical knockoffs due to the transformation $\gamma = D\beta$. Thus, standard exchangeability does not hold, making the Lasso coefficient difference statistics invalid. Instead, our lifted construction only requires the weaker directional property $\mathbb{P}(W_l\le 0)\ge 1/2$ for null features, which is sufficient for FDR control (see [1] and Lemma B.1 in our paper).
>
> [1] Barber & Candès (2019). A knockoff filter for high-dimensional selective inference.
>
> Question. About empirical covariance in Eq.4.
>
> Response. We would clarify that we consider the fixed-design setting, where the knockoff copies are designed to mimic the original design in terms of the empirical covariance matrix, rather than the randomized design, where we should consider the population covariance matrix when generating knockoff copies.
>
> **Question.** More simulation experiments and SNR specification.
>
> Response.  In the paper, we set the noise level to $\sigma = 1$. We will clarify it in the revised version.
>
> We conducted additional simulations under high correlation, alternative $D$, high-dimensional screening, and lower SNR. Representative FDR/power results are $0.11/1.00$ and $0.16/0.98$ for fused $D_G$ with $c=0.8,0.9$, $0.00/1.00$ for new $D=[I,D_G^\top]^\top$, $0.23/0.67$ in the high-dimensional fused setting, and $0.12/1.00$, $0.11/1.00$ for low SNR $(A,\sigma)=(7,1.5)$ and $(5,2)$, respectively. These results support robustness while controlling FDR $(q=0.2)$ in all settings.  Due to the response length limit, we refer the reviewer to our response to Reviewer be2v for details.
>
> **Question.** Choice of $\nu$ and sample splitting.
>
> Response. We clarify that $\nu$ does not affect FDR control, since Theorem 3.1 guarantees it uniformly for all $\nu > 0$; this is also supported by Figure 2. The role of $\nu$ is instead to only affect power, as discussed after Proposition 3.5.
>
> Regarding the sample-splitting strategy, we agree that it may entail some loss of efficiency. However, this step is needed in our analysis to ensure that the elements of $\zeta$ in Eq.18 are mutually uncorrelated, which is crucial for establishing Lemma B.2, and hence for proving FDR control.
>
> **Question.** About the neuroimaging experiment
>
> Response. In Table 2, the reported values represent the selection frequency over 10 runs. A higher frequency indicates that the brain region is more likely to be identified as significant. We will clarify this in the revision.
>
> We pool 1.5T and 3T measurements because the inferential target is the brain region, not the scanner-specific effect. Although 3T may be more informative, this affects the evidence strength within a group rather than the group-level target itself.
>
> Different splitting ratios are used because $D$ has different dimensions in region and connection selection, and the knockoff step must ensure $n_2>m+p$.
>
> **Question.** SGK vs. SGK+ and power at small $q$.
>
> Response. The defination of “+” and non-“+” follows the original knockoff[1] and is standard in the subsequent literature. SGK+ provides exact FDR control but is more conservative, especially when the support is small or $q$ is low. By contrast, SGK is typically more powerful at the cost of controlling a slightly modified FDR. This trade-off is widely recognized in knockoff literature. In practice, it is therefore standard to report both versions.
>
> [1] Barber & Candès (2015). Controlling the false discovery rate via knockoffs.

---

> > ### Author Rebuttal · Reviewer_hSWt · 2026-04-01
> >
> > I understand that the Lasso Coefficients cannot be used, but group lasso coefficients ?
> >
> > Otherwise, I think the authors responses acknowledge the problems rather than addressing them.

---

> > > ### Author Response · Authors · 2026-04-02
> > >
> > > **Response.**
> > >
> > > We thank the reviewer for the follow-up. We would like to clarify that the issue is **not** whether one uses Lasso or group Lasso. The difficulty comes from the **transformed sparsity target** $\gamma = D\beta$. Because of this transformation, the resulting construction is generally **non-symmetric**, so the usual coefficient-difference type statistics do not have the required validity property.
> > >
> > > We respectfully disagree with the characterization that we merely acknowledge the problem. The non-exchangeability of the (group) lasso coefficient is exactly why it cannot be used directly. Our contribution is to replace that invalid quantity with a new knockoff-based statistic $W_i =Z_i \cdot \mathrm{sign}(Z_i - \widetilde{Z}_i)$, built from paired original/knockoff importance scores. This statistic is designed to satisfy the weak directional property needed for knockoff thresholding, which allows us to recover rigorous FDR control. Therefore, the non-exchangeability  issue of the (group) LCD is not simply noted as a limitation; it is the technical obstacle that our proposed construction explicitly overcomes.

---

### Official Review · Reviewer_be2v · 2026-03-13

**Soundness:** 3
**Presentation:** 3
**Significance:** 3
**Originality:** 3
**Overall Recommendation:** 4
**Confidence:** 3

**Summary:**

The paper proposes an extension for the Split Knockoff procedure to work for grouped transformed variables. They apply both theoretical guarantees and empirical studies to show that Split Group Knockoff can identify these group-level signals with false discovery rate control. Evaluation is done over both general auditing tasks and specific Alzheimer's related tasks.

**Compliance With Llm Reviewing Policy:**

Affirmed.

**Final Justification:**

My concerns have been mostly addressed by the rebuttal and I maintain my positive assessment of this paper. I believe it provides a solid contribution to the field as a start to looking at split group knockoff procedures.

**Key Questions For Authors:**

1. Is it possible to compare SGK with some standard group methods on the transformed design matrix?
2. What are the computational costs of SGK like when looking at high dimensional methods?
3. The numerical experiments seem somewhat limited, would be interesting to see some more options for D.

**Limitations:**

Yes

**Strengths And Weaknesses:**

**Soundness**

Strengths:
- Solid extension of the split knockoff framework to grouped transformer variables
- Synthetic numerical experiments provide good support for the FDR control

Weaknesses:
- No comparison to any existing baselines

**Presentation**

Strengths:
- Well-structured

Weaknesses:
- Method is a little confusing to follow

**Significance**

Strengths:
- Addresses a gap in existing group methods that does not look at sparsity after transformations
- Potential applications across many fields (including behavior auditing, neuroscience. etc)

Weaknesses:
- Not sure how this could be practically utilized

**Originality**
- New problem formulation and theoretical extension upon existing split knockoff

---

> ### Author Rebuttal · Authors · 2026-03-30
>
> We thank the reviewer for the thoughtful feedback and helpful questions. We address the concerns below.
>
> **Question.** Comparison to other baselines.
>
> **Response.** We thank the reviewer for this comment. To the best of our knowledge, the transformational group sparsity setting has not been considered in prior literature on selective inference. Specifically, previous methods controlled FDR in either the transformational sparsity setting [1] or the standard group sparsity setting [2], and these methods cannot apply to our setting.
>
> [1] Cao, Y., Sun, X., & Yao, Y. (2024). Controlling the false discovery rate in transformational sparsity: Split knockoffs.
>
> [2] Dai, R., & Barber, R. (2016). The knockoff filter for FDR control in group-sparse and multitask regression.
>
> **Question.** Computational cost.
>
> **Response.** We thank the reviewer for raising this point. The computational cost of the high-dimensional SGK procedure remains at a reasonable level. To demonstrate, we calculated the running time in the high dimensional setting simulation, and the result is 34 seconds for cv beta and 87 seconds for path in one repetition, while the standard knockoff in low dimensional setting. Overall, this suggests that the high-dimensional extension introduces only a modest overhead and remains computationally practical.
>
> **Question.** Experimental validations in more settings.
>
> **Response.**  In addition to the settings in the paper, we conduct experiments under (i) high correlation ($c \in \{0.8, 0.9\}$), (ii) alternative transformations, where $D=[I,D_G^\top]^\top$ combines the identity matrix and the 1-d fused lasso matrix, (iii) high-dimensional settings with screening, and (iv) lower-SNR settings. We report representative results below, using the same default setup as in the manuscript unless otherwise specified.
>
> | Setting   | c   | (A, $\sigma$) | D        | Method  | $\log_{10}(\nu)$ | FDR    | Power  |
> |----------|-----|--------|----------|---------|----------|--------|--------|
> | High corr | 0.8 | (10,1) | fused    | cv-beta | 2.6      | 0.1652 | 1.0000 |
> | High corr | 0.8 | (10,1) | fused    | path    | 2.0      | 0.1095 | 1.0000 |
> | High corr | 0.9 | (10,1) | fused    | cv-beta | 2.6      | 0.1602 | 0.9800 |
> | High corr | 0.9 | (10,1) | fused    | path    | 2.0      | 0.1301 | 1.0000 |
> | Alt $D$   | 0.5 | (10,1) | combined | cv-beta | 2.0      | 0.0049 | 1.0000 |
> | Alt $D$   | 0.5 | (10,1) | combined | path    | 2.0      | 0.0774 | 1.0000 |
> | High-dim  | 0.5 | (10,1) | identity | cv-beta | 1.0      | 0.0000 | 0.9900 |
> | High-dim  | 0.5 | (10,1) | identity | path    | 1.0      | 0.0000 | 1.0000 |
> | High-dim  | 0.5 | (10,1) | fused    | cv-beta | 1.8      | 0.2444 | 0.6200 |
> | High-dim  | 0.5 | (10,1) | fused    | path    | 1.8      | 0.2338 | 0.6700 |
> | SNR       | 0.5 | (7,1.5)| fused    | cv-beta | 1.8      | 0.1524 | 1.0000 |
> | SNR       | 0.5 | (7,1.5)| fused    | path    | 1.0      | 0.1219 | 1.0000 |
> | SNR       | 0.5 | (5,2)  | fused    | cv-beta | 1.0      | 0.1751 | 0.9950 |
> | SNR       | 0.5 | (5,2)  | fused    | path    | 1.0      | 0.1100 | 1.0000 |
>
> The results show that our methods can consistently control FDR across these settings.

---

> > ### Author Rebuttal · Reviewer_be2v · 2026-04-03
> >
> > Thanks for the response, my concerns have been mostly addressed.

---

### Decision · Program_Chairs · 2026-04-30

**Decision:**

Accept (regular)

**Comment:**

Most reviews contain positive statements about the clarity of the presentation, the novelty of the transformational group sparsity concept,  and interesting experiments / applications. On the other hand, several critical concern have been raised as well, such as:
- the use of unverifiable assumptions in the theoretical analysis;
- absence of strong baseline methods for comparison;
- limited significance due to the assumption that the labels come from a linear model,
- not fully convincing over-all motivation.
Some of these concerns could be addressed in the rebuttal in a convincing way, others not (the more conceptual ones), so this still seems to be a borderline case. After carefully re-reading all reviews rebuttals and comments again, however, I think that the underlying idea is indeed interesting and potentially inspiring for the ML community. For this work, I think that finally the conceptual novelty outweighs the (obvious) weaknesses, and therefore I recommend (weak) acceptance (although I'd like to re-iterate that this is still a somewhat controversial paper).